

# 1  Understanding microbial sourcing in Greenland subglacial
# 2  runoff

Guillaume Lamarche-Gagnon[1,2*], Marek Stibal[3], Alexandre M. Anesio[4], Jemma L.
Wadham[1,2], Jon R. Hawkings[1,5], Lukáš Falteisek[3], Kristýna Vrbická[3], Petra Klímová[3], Jakub
D. Žárský[3], Tyler J. Kohler[3], Elizabeth A. Bagshaw[2], Jade E. Hatton[3], Alexander D. Beaton[6],
Jon Telling[7]
[1]Centre for ice, Cryosphere, Carbon and Climate (iC3), Department of Geosciences, UiT The Arctic University
of Norway, 9037 Tromsø, Norway
[2]School of Geographical Sciences, University of Bristol, Bristol BS8 1SS, UK
[3]Department of Ecology, Faculty of Science, Charles University, Prague, 128 44, Czechia
[4]Department of Environmental Science, Aarhus University, Aarhus, 4000 Roskilde, Denmark
[5]Department of Earth and Environment, University of Pennsylvania, Philadelphia, PA, USA
[6]Ocean Technology and Engineering Group, National Oceanography Centre, Southampton, UK
[7]School of Natural and Environmental Sciences, Newcastle University, Newcastle, UK
*Correspondance to: Guillaume Lamarche-Gagnon (guillaume.lamarche-gagnon@uit.no)
**Abstract.** The microbial ecosystems that lie beneath ice sheets can impact and contribute to global
biogeochemical cycles, yet remain poorly understood given the logistical challenges in directly accessing the
subglacial environment. Studies instead often rely on indirect sampling of subglacial systems via the collection of
meltwaters emerging from ice margins. However, the origin of exported material in these waters will change over
a melt season as glacier hydrology responds to changes in surface melt. Here, we reveal trends in microbial
sourcing (source environment) and assemblages in a large proglacial river in southwest Greenland by investigating
three microbial datasets (16S rRNA) collected during different hydrological periods over three separate summer
melt seasons. By combining microbial data with high-resolution hydrological and hydrochemical measurements,
we show that changes in microbial assemblages follow changes in hydrological periods, likely influenced by
variations in glacial drainage expansion inland with concomittant variations in inputs of surface melt and
subglacial sediment exports. We further illustrate how relative changes in microbial assemblages can inform on
the state of the glacial hydrological system, and also focus on methane-cycling populations to infer their potential
distribution beneath the ice. Overall, our results highlight that timing matters when sampling proglacial rivers and
we caution interpretations of exported assemblages without a good understanding of the catchment and system
studied; this is especially true for larger systems which undergo more complex hydrological changes over a melt
season.

## 32  1 Introduction

The beds of glaciers and ice sheets are now recognised as widespread ecosystems conducive to biochemical and
biogeochemical reactions with impacts well beyond the ice margins (Wadham et al., 2019; Christner et al., 2014).
Subglacial ecosystems promote chemical weathering reactions, leading to the potential release of nutrients (e.g.
Si, Fe) and heavy metals (e.g. Hg) to proglacial zones and nearby fjords and oceans (e.g. Graly et al., 2017;
Hawkings et al., 2021b; Hawkings et al., 2014), and the generation and build-up of subglacial greenhouse gas
reserves (methane) that can leak to the atmosphere (e.g. Burns et al., 2018; Lamarche-Gagnon et al., 2019;
Christiansen and Jørgensen, 2018; Pain et al., 2019). Despite this importance, our understanding of subglacial



microbial processes and communites remains limited, hindered by the difficulty to directly access the subglacial
environment.

An increasing number of studies have circumvented this logistical challenge by sampling subglacial material in
meltwaters exported from the ice margin. In most glacial systems, such as those found in the ablation zone of the
Greenland Ice Sheet (GrIS), surface melt reaches the bed of the ice sheet before exiting glaciers at their front; for
land-terminating glaciers, glacial runoff ultimately takes the form of proglacial rivers. During the melt season,
these rivers can therefore offer indirect access to the subglacial environment, and concomitantly to the microbial
habitats it encompasses (e.g. Dieser et al., 2014; Kohler et al., 2020; Cameron et al., 2017; Žárský et al., 2018;
Dubnick et al., 2017). However, the origin of exported material changes over an ablation season as increasing
surface melt re-organises subglacial drainage pathways following snowline retreats on the glacier surface.

Glaciers with temperate ice regions in Greenland typically undergo a transition from slow and inefficient
subglacial drainage (distributed system) during the early part of the melt season, with meltwater during this period
normally sourced from sectors of the ice sheet closer to the margin, to efficient fast-flow subglacial drainage in
later months draining a larger catchment area and more remote inland sectors of the bed (channelised system;
Röthlisberger, 1972; Weertman, 1972; Davison et al., 2019). The degree of mixing between surface melt and the
subglacial environment subsequently varies, influencing the quantity and the origin of subglacial material that is
transported to the ice margin. Proglacial rivers are consequently sourced from waters of varying residence time
beneath the ice and from different sectors of the bed, depending on the state of the hydrological system, which
makes disentangling the origin of exported material challenging, and often speculative. Timing of sampling can
therefore influence, and potentially skew, interpretations if no additional information on the state of the glacier's
hydrological system is considered. This is especially true for larger glacial systems, which undergo more dramatic,
pronounced and complex hydrological change throughout the melt season, draining more expansive areas, and
likely exporting older, more isolated bed material to the proglacial zone as meltwater generation increases
(Wadham et al., 2010; Kohler et al., 2017).

Here, we provide a detailed investigation into the microbiome of the proglacial river draining the Leverett Glacier
(LG), a very well studied large land-terminating glacier of the southwestern sector of the GrIS. We evaluate
changes in microbial assemblages (16S rRNA gene) exported from beneath the catchment as the glacier undergoes
changes in hydrological connectivity during the first half of the 2015 melt season (e.g. Lamarche-Gagnon et al.,
2019; Hatton et al., 2019; Kohler et al., 2017), and also re-visit data from 2017 during late-melt as well as in early
to mid summer 2018 before peak flow (Vrbická et al., 2022; Kohler et al., 2020), in order to capture inter-annual
variability and a wider range of hydrological periods over three separate summer melt seasons. We assess observed
changes based on combined biogeochemical, hydrological, and microbiological data to qualitatively gauge the
degree of mixing and homogenisation between different components of the drainage system (e.g. from waters of
different residence time subglacially) occurring during fluvial transport upstream of the sampling site.
Futhermore, we evaluate whether changes in exported assemblages might point to the existence of different
habitats or geochemical conditions in the subglacial catchment and demonstrate that microbiological data can
supplement more traditional geochemical and hydrological data to better understand the glacial hydrological



system during certain sampling periods. Given the increased attention surrouding glacial emissions of microbial
methane from this sector of the ice-sheet (e.g. Stibal et al., 2012; Lamarche-Gagnon et al., 2019; Christiansen and
Jørgensen, 2018; Pain et al., 2019; Dieser et al., 2014), we further apply these interpretations to focus on putative
methane-cycling populations to see what additional information can be gained about methane cycling beneath the
GrIS.
**2 Methods**
**2.1 Site description**
The hydrology and hydrochemistry of Leverett Glacier (LG) have been extensively documented over the last
decade (Clason et al., 2015; Chandler et al., 2013; Bartholomew et al., 2011; Hawkings et al., 2021b; Kohler et
al., 2017; Hawkings et al., 2014). LG is a polythermal, land-terminating glacier draining a subglacial catchment
of ~600-1200 km$^2$ in the southwestern margin of the GrIS (Cowton et al., 2012; Palmer et al., 2011; Hawkings et
al., 2021b). LG overlies Precambrian orthogneiss and granite, common lithology to much of Greenland (Dawes,
2009), and its proglacial river, sampled here, is the main source of the Akuliarusiarsuup Kuua, itself a tributary,
alongside the Qinnguata Kuussua, to the Watson River near the settlement of Kangerlussuaq. Over the course of
a melt season, the catchment undergoes spatial hydrological expansion with processes characteristic of the western
margin of the GrIS (details in the Results section below; Davison et al., 2019; Chandler et al., 2013).
**2.2 Hydrological and hydrochemical data**
All hydrological sensor and hydrochemical data presented here have been described in detail elsewhere (Hawkings
et al., 2021b; Kohler et al., 2017). Briefly, continuous measurements of water discharge and suspended sediment
concentrations were achieved via the deployment of hydrological sensors for stage (HOBO Onset or Keller DCX-
22-CTD) and turbidity (Partech C connected to Campbell CR1000 logger or Turner Cyclops-7F connected to
Campbell CR1000 logger) over the span of the 2015 and 2018 sampling periods. Stage was converted to discharge
via calibration against a rating curve constructed from manual rhodamine dye injections; turbidity was converted
to suspended sediment concentrations (SSC) via calibration against manual samples filtered on pre-weighed
cellulose nitrate filters. In 2015, the discharge record at LG spanned the entirety of the melt season, allowing to
better contextualise the microbiological sampling period relative to the rest of the melt season. To allow for similar
comparisons for the 2017 samples and the 2018 sampling period, we also include complete discharge records
from the Watson River (Van As, 2022), of which the LG river is a main tributary (~25 – 65 % of overall Watson
daily discharge for the overlapping measurement period), and consequently discharge between the two sites follow
very similar patterns (Fig. 1).

Hydrochemical data collected in parallel to microbiological samples have also been described elsewhere
(Hawkings et al., 2021b; Kohler et al., 2020; Kellerman et al., 2020; Hatton et al., 2019; Lamarche-Gagnon et al.,
2019; Kohler et al., 2017). Here, we also include dissolved silica (DSi), major ion (MI) and dissolved methane
(CH$_{4(aq)}$) concentrations, as well as dissolved oxygen, pH, and electrical conductivity (EC) data for the early May
2015 samples collected via ice borehole and chainsawed holes directly in front of the LG terminus (<10 m) prior
to glacial melt onset (Table S1; Lamarche-Gagnon et al., 2019). Borehole water samples for DSi, MI and CH$_{4(aq)}$



anlayses were first collected using a peristaltic pump (Portapump-810, Williamson Manufacturing) equipped with
pre-autoclaved silicon tubing extensively flushed with sample water (> 2 L) prior to sample collection. $CH_{4(aq)}$
samples were directly added to pre-evacuated borosilicate vials and analysed as described in Lamarche-Gagnon
et al. (2019); subsamples for DSi and major ions were kept cold and then filtered through 0.45 µm cellulose nitrate
membrane filters within 2 to 5 hours of collection and analysed as per Hatton et al. (2019). Dissolved oxygen
(DO), pH, and EC were only measured for water collected on 4 May 2015 by deploying hydrological sensors
through the borehole at ~3-5 meters beneath the river-ice surface; sensors (Aanderaa 3830 for DO, Campbell 247
for EC, and Honeywell Durafet for pH) are described in Beaton et al. (2017).
**2.3 Microbiological sampling**
Except for three samples collected beneath or on river ice in early May and June 2015, respectively (see below),
all samples consisted of LG bulk proglacial runoff collected from the river bank. Due to logistical reasons, water
volumes and collection sites alongside the LG proglacial river differed slightly between sampling periods and
melt seasons but were broadly within 2 km of the glacier terminus (Fig S1, Table S2). In 2015, the sampling
period spanned from 4 May to 26 July 2015, with most samples collected ~500 m from the ice margin, except for
the first four sets of samples. The first three sets of samples in early May (4-13 May 2015) consisted of relatively
stagnant waters collected through naled river ice accessed via a borehole and a chainsawed hole as described in
the above section. The following set of samples (7 June) were also collected near the glacier's margin (<20 m)
onto river ice, but this time consisted of flowing waters emerging through the river ice in the form of an upwelling
that fed the proglacial river at that time. The remaining of the 2015 samples (n=41) were collected from the river
bank < 500 m from the ice margin.  In 2017, samples (n=3) were collected during a single day during the late melt
season near the ice margin (5 September 2017; Kohler et al., 2020). In 2018, the sampling period spanned from
21 June to 13 July, with samples collected ~ 2 km from the ice margin (Vrbická et al., 2022).
All water samples were filtered onto Sterivex filters (0.22 µm, Millipore, Billerica, MA, United States), collected
either using sterile 60 mL plastic syringes (2017 and 2018 and most 2015 samples, n = 45) or a peristaltic pump
(as described above; 2015 samples, n = 13). Sterivex filters were preserved in MoBio RNA LifeGuard solution
(MoBio Laboratories, USA) immediately after sampling and frozen inside a portable freezer (<-10°C) within 1
hour of collection. Most water samples collected in 2018 consisted of a single sterivex filter per sampling event,
and all 2017 and most 2015 samples were collected in replicates (Table 1/S2). Details on sampling time, location,
methods, filtered water volume (usuallly filtered until clogging), and sample replicates are summarised in Table
S2.
Importantly, we maintain that these minor differences in sampling locations and sample processing have little to
no measurable impact on microbial composition given that LG subglacial runoff by far consitutes the majority of
the discharge of the proglacial river sampled. No major tributary to the LG proglacial river exists within the first
2 km of the river, and the area of the catchment that is non-glaciated (~ 11 km2; Vrbická et al., 2022) is orders of
magnitude smaller than the overall LG glacial catchment sourcing the proglacial river (~ 840 km2; Hawkings et
al., 2021b). Moreover, the relatively fast flowing waters (~1 m s$^{-1}$) and substantial discharge (often >100s m$^3$ s$^{-1}$;
Fig. 1) would not allow for any meaningful ecological processes to occur within the first 2 km of river sampled.



**2.4 DNA extraction and sequencing**

DNA from all samples was extracted using the PowerWater Sterivex DNA isolation kit (MO BIO) following the manufactuer's protocol; the 2015 DNA extracts were further purified using the Genomic DNA Clean & Concentrator (Zymo Research, Irvine, CA, USA) as in Žárský et al. (2018) (Lamarche-Gagnon et al., 2019; Vrbická et al., 2022; Kohler et al., 2020). In 2015, extracted DNA samples were then pooled into triplicates for dates when more than 3 Sterivex water samples were collected. For 3 sets of samples, DNA extracts from different (but consecutive) time points were pooled into triplicates (20-23 June, 13-17 July, 21-26 July 2015); Tables 1 and S2 summarises grouping and pooling of the different DNA samples prior to sequencing.

Amplification and sequencing for all samples targeted the V4 region (515-806) of the 16S rRNA gene and performed on a an Illumina MiSeq platform (2 X 250 bp). 2015 and 2017 DNA samples were sequenced at the Mr. DNA laboratory (Shallowater, TX, USA) and amplified using the Caporaso et al. (2011) 515F (GTG**C**CAGCMGCCGCGGTAA) and 806R (GGACTAC**H**VGGGTWTCTAAT) primer pair. 2018 samples were sequenced at SEQme s.r.o. (Dobříš, Czechia) but using the slightly modified Parada et al. (2016) 515F-Y (GTG**Y**CAGCMGCCGCGGTAA) and Apprill et al. (2015) 806R (GGACTAC**N**VGGGTWTCTAAT ) primer pair. We acknowledge that the comparison of separate datasets generated using different primer pairs can introduce a degree of primer bias. However, we consider that the difference in primer pairs here should only have a minor impact on the overall sequencing results given the Parada and Apprill primer pair was designed to improve the detection of SAR11 marine taxa and freshwater bacterioplanktons, as well as Crenarchaeota/Thaumarchaeota, clades not expected to make up a large component, if at all, of the subglacial microbial communities investigated here (e.g. Vrbická et al., 2022; Cameron et al., 2017; Henson et al., 2018).

**2.5 Bioinformatics analyses**

A pre-selection of samples was first made based on a prior screening of the 2015 and 2018 datasets to exclude samples with relatively low numbers of reads per sample. This resulted in an initial sample set (including replicates) of 31 samples for 2015, 3 samples for 2017, and 13 samples for 2018 (compared to 29 LG samples included in Vrbická et al. (2022)) before bioinformactics processing (final processed sample numbers below).

Raw sequences were then uploaded on a remote server and analysed using the mothur platform for preprossessing (Linux v.1.42.3; Schloss et al., 2009). Paired reads were first converted into contigs for each dataset separately, before being merged into a single dataset (*see supplementary information*). Analyses then mostly followed the mothur MiSeq standard operating procedure performed in "batch mode" (https://mothur.org/wiki/miseq_sop/; Kozich et al., 2013). In short, sequences were binned into operational taxonomical units (OTUs) at a 97% sequence identity level using the OptiClust method (Westcott and Schloss, 2017) and classified against the SILVA (v.132) database (Quast et al., 2012), following quality and chimera checks (uchime v 4.2 http://www.drive5.com/uchime). Sequences related to chloroplasts, mitochondria, Eukaryota, or classified as 'unknown' (at the domain level) were also removed.

Downstream analyses (alpha/beta diversity) were performed on a local machine (macOS; mothur v1.48). Samples were sub-sampled down to 40 206 reads either before or during diversity analyses (see below), resulting in the



exclusion of 5 samples from the 2018 dataset; the final dataset therefore contained 31 samples from 2015, 3 from
2017 and 8 from 2018 (including replicates; Table 1). Both measures of alpha and beta (Bray-Curtis dissimilarity)
diversity were calculated with default settings; averages from sub-sampling iterations (1000) were used in both
cases. Distances in Bray-Curtis dissimilarities were visualised as ordinations using principal coordinate analyses
(PCoA) and non-metric multidimensional scaling (nMDS) calculated in mothur. Full details on the specific
mothur commands used are provided as Supplementary Information.
**2.6 LEfSe analyses**
One aim of the present study was to assess potential changes in microbial assemblages exported in proglacial
runoff over different stages of melt during the ablation season as the hydrological drainage system of LG
undergoes re-configuration (e.g. distributed to efficient). To identify which major OTUs (LDA scores > 3.5, p-
value < 0.01) were over- or under- represented during six different hydrological periods at LG (see Results section
below), we used linear discriminant analysis effect size (LefSe) implemented in mothur (v.1.44.3). Relative
changes in relative abundance across all samples for the identified OTUs were then visualised as a heatmap.
**2.7 Identification of methane-cycling populations**
A detailed characterisation of methane-cycling communities at LG is beyond the scope of the present study.
However, major clades of both methanotrophic and methanogenic populations from subglacial outflows in the
same GrIS sector have been described previously (Vrbická et al., 2022; Lamarche-Gagnon et al., 2019; Dieser et
al., 2014; Stibal et al., 2012; Znamínko et al., 2023). Based on these previous reports, we herein define putative
methanotroph bacterial populations as OTUs of the genera *Methylobacter* and *Crenothrix* (Znamínko et al., 2023),
and putative methanogen populations as OTUs of the orders Methanobacteriales, Methanomicrobiales,
Methanosarcinales (Lamarche-Gagnon et al., 2019). Screening for methanogenic populations also revealed a
relatively high abundance of reads classified as the anaerobic methane-oxidising archaea *Candidatus*
*methanoperedens* (Cai et al., 2018) amongst the Methanosarcinales in some samples. OTUs classified as *C.*
*methanoperedens* were therefore also included as potential methane-oxidising archaea.
**2.8 Data manipulation and visualisation**
Unless stated otherwise, all data manipulation and visualisation were performed in R (v. 4.2.1, R Core Team,
2013) utilised through RStudio (v. 2023.06.2, R Studio Team, 2020) using packages from the Tidyverse
(Wickham et al., 2019).





**Table 1: Summary of number of water (Sterivex) and bioinformatics samples**

| Melt season | Time Span | Sterivex Samples | Bioinfo. Samples | Bioinfo. Replicate | Time Points* | Hydrological Period | Primer Pair** | Dataset |
|---|---|---|---|---|---|---|---|---|
| 2015 | 4 – 13 May | 4 | 3 | 1 | 3 | borehole | Caporaso 2011 | Lamarche-Gagnon et al. 2019 |
| 2015 | 7 June 2015 | 2 | 2 | 2 | 1 | early-melt | Caporaso 2011 | Lamarche-Gagnon et al. 2019 |
| 2015 | 20 June – 10 July | 30 | 20 | 2-3 | 7 | outburst | Caporaso 2011 | Lamarche-Gagnon et al. 2019 |
| 2015 | 13-26 July | 11 | 6 | 1-3 | 3 | peak-flow | Caporaso 2011 | Lamarche-Gagnon et al. 2019 |
| 2017 | 5 Sept | 3 | 3 | 3 | 1 | late-melt | Caporaso 2011 | Kohler et al. 2020 |
| 2018 | 24 June – 24 July | 8 | 8 | 1-2 | 7 | transition | Parada-Apprill 2016-15 | Vrbická et al. 2022 |

Sterivex samples correspond to the number of Sterivex filters collected and subjected to DNA extraction. Bioinformatics samples
correspond to the number of DNA samples sequenced. Bioinformatics replicates correspond to the number of replicate samples
for each group of samples. See Table S2 for details.
*In 2015, samples were collected on 17 separate days, but three sets of samples were pooled into single samples prior to
sequencing resulting in 14 sets of samples. The 2018 dataset here contains one sample per day except for 25 June 2018, which
was sampled twice at ~3 hr intervals (see methods/Table S2).
***The V4 (515-806) primer pair used during Illumina MiSeq sequencing. Caporaso 2011 refers to the Caporaso et al. (2011)
primer pair and the Parada-Apprill 2016-15 refers to the Parada et al. (2016) 515F-Y and Apprill et al. (2015) 806R primer pair
(see methods).



## 3 Results

### 3.1 The hydrological evolution of the LG proglacial river (water sourcing)

The hydrology and hydrochemistry of the LG river reflect the status of its drainage system and can therefore inform on the sourcing of proglacial waters and their suspended material. To interpret changes in sampled microbial assemblage structure within the context of their upstream sources beneath or over the ice, we re-visit previously described hydrological and hydrochemical data at LG (Kohler et al., 2017; Hatton et al., 2019; Hawkings et al., 2021b). Figure 1 puts microbiological sampling in the context of the hydrological status of LG based on our understanding of its drainage system. Based on changes in runoff hydrochemistry (e.g. Hatton *et al.* 2019), flow regime, and SSC, we separate microbiological samples into six different "hydrological periods" or categories between the three years of sampling:

1. Boreholes: In 2015, the first set of samples were collected prior to the melt onset and consisted of relatively stagnant water accessed via boreholes through naled river-ice (see methods) and hereafter referred to as "2015 borehole" samples (4-13 May 2015; n = 3). This set of samples were the only samples that did not consist of flowing waters (i.e. runoff), but most likely consisted of a mixture of over-wintered late-runoff from the previous year (i.e. storage waters) and basal meltwaters (e.g. Graly and Rezvanbehbahani, 2023). These waters contained relatively high methane concentrations ($\sim 4\ \mu M$), were hypoxic (25.8 $\mu M$ or < 6% air saturation) and contained very little suspended sediment but overall higher dissolved ion concentrations than bulk runoff (Table S1; Hatton et al. (2019)).

2. Early melt period: The second set of 2015 samples consisted of runoff samples collected during the early melt season (7 June 2015; n = 2). At this point in the season, the glacial hydrological system was poorly developed, and most waters exiting the glacier consisted of over winter stored basal meltwaters from/near the margin of the glacier (Bartholomew et al., 2011; Kellerman et al., 2020). These waters were more diluted by surface and englacial melt than the borehole samples, but still showed low SSC and relatively high EC, solute and $CH_4$ concentrations compared to the rest of the of the runoff samples, indicative of less dilution by surface melt than later in the season (Lamarche-Gagnon et al., 2019; Hatton et al., 2019).

3. Outburst period: The highest number of microbial samples in 2015 spanned a period where the LG hydrological drainage systems was undergoing rapid spatial expansion following the "spring event" (e.g. Mair et al., 2003), between 19 June – 15 July 2015 ("2015 outburst" samples; n = 20). This period has been extensively described in previous studies (Hawkings et al., 2018; Lamarche-Gagnon et al., 2019; Hatton et al., 2019; Kohler et al., 2020; Kellerman et al., 2020). In short, the series of four pulses (outbursts) in Q and SSC (Fig. 1) during that period reflect the rapid drainage of supraglacial waters to the base of the glacier (Bartholomew et al., 2011). This large input of dilute surface meltwater mechanically disrupts the glacier's subglacial hydrology, flushing out waters with high sediment (and $CH_4$) loads from newly connected sectors of the glacier's bed (Bartholomew et al., 2011; Lamarche-Gagnon et al., 2019).



4. Peak flow period: The last set of of microbial samples in 2015 were collected directly following the outburst period (15 – 23 July) during peak flow ("2015 peak flow" samples; n = 6). The drainage system then experienced full spatial extent where surface meltwaters quickly transit through efficient subglacial drainage conduits before exiting the ice, reflected by the drop in SSC, relatively low EC, and high Q (Fig. 1; Hatton et al., 2019; Chandler et al., 2021; Chandler et al., 2013).

5. Late-melt period: The 2017 samples ("2017 late-melt") were collected towards the end of the melt season (Kohler et al., 2020). Whilst no continuous hydrological LG data is available for 2017, the Watson River hydrograph indicates that the LG samples were collected just prior to the shutdown of the LG hydrological system during a time of flow recession (Fig. 1). Runoff at this time is inferred to mostly flow through pre-existing channels developed earlier during the melt season (Davison et al., 2019).

6. 2018 transition period: The 2018 set of samples were collected during the same time period as the "outburst period" in 2015 (24 June to 11 July 2018; Vrbická et al., 2022), a few weeks following the putative spring event (7 June) but before peak flow (22 July; Fig. 1). However, no clear indication of outburst hydrological pulses occurred during the sampling period (Hawkings et al., 2021b; Vrbická et al., 2022). While Q decreased over the first week in June before increasing steadily during the July period, SSC decreased over the entire sampling period (Fig. 1). Because no distinctive hydrological or hydrochemical events characterised the 2018 sampling period, we group all of the microbial samples collected in 2018 into a single hydrological period: "2018 transition period"; n = 8.

The number of microbiological samples from each dataset and hydrological period is summarised in Tables 1 and S2.

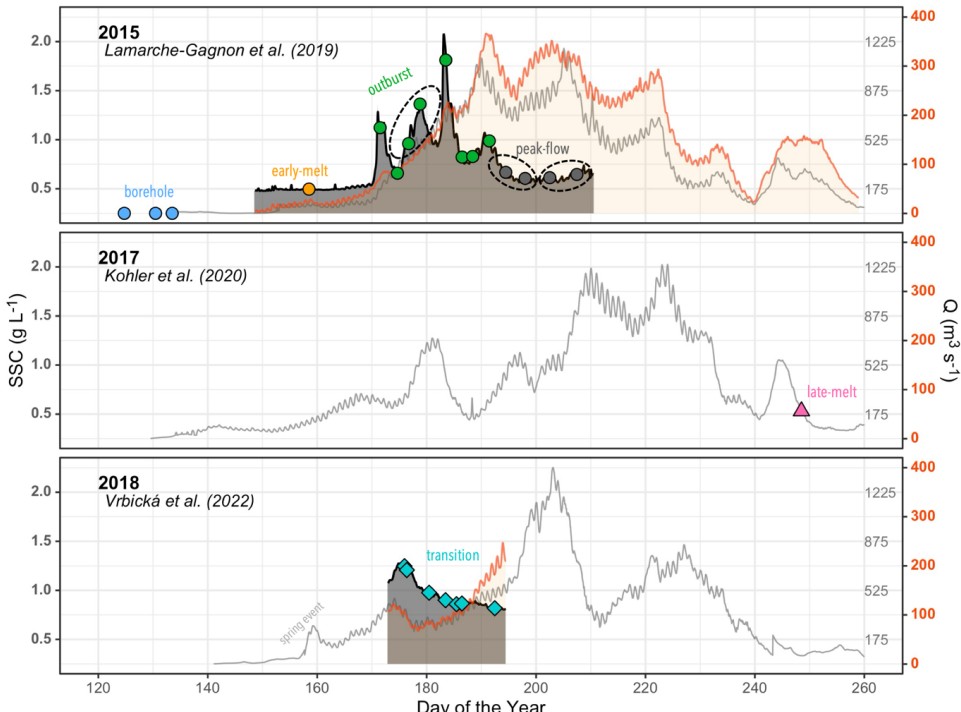

**Figure 1: Hydrological and hydrogeochemial evolution of the LG proglacial river for the 3 years of sampling.** Dark-grey and orange ribbon timeseries respectively correspond to suspended sediment concentration (SSC) and discharge (Q) at LG; no Q or SSC data in 2017 and limited data in 2018. Light grey line timeseries correspond to the Watson River discharge measured ~25 km downstream (data from Van As, 2022). Coloured points overlayed onto the SSC (2015, 2018) or Watson River Q ( 2017) timeseries indicate the sampling time of waters used for DNA extraction (Sterivex filters). Fill colours correspond to the six different hydrological periods identified (see Results). In 2015, dashed elipses highlight samples that have been merged from different sampling days prior to sequencing (see Table 1, S2). References for the original microbial datasets used are identified beneath the melt season year.

### 3.2 Microbial diversity and composition

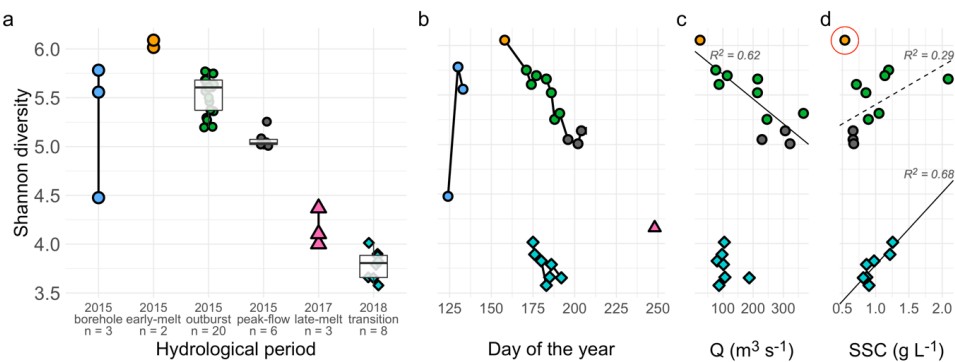

**Figure 2: Shannon alpha diversity.** (a) Shannon alpha diversity separated by hydrological period. All replicate samples are displayed; when n > 3, boxplots displays median, interquartile range (IQR; box) and 1.5 X IQR (whiskers). (b) Shannon diversity versus day of the year. Averages between replicates are shown with range displayed as vertical error bar (when larger





than point). Horizontal error bars are only barely visible for the last 2015 data point, and present for sample points derived
from multiple sampling days (see methods; Table S2). Points of the same dataset are linked by black line; for the 2015 dataset,
borehole (blue) and runoff samples (rest) are separated. (c) Shannon diversity versus discharge (Q) and (d) suspended sediment
concetration (SSC) for runoff samples; averages of replicates displayed. In c-d, solid black line display significant (p < 0.05)
correlations whereas the dashed line in (d) displays a weak, though non-significant, correlation in 2015 when exluding the
early-melt samples (orange point, circled in red). Same y-axis, fill colour and shape scheme is used in all panels.

In terms of microbial alpha-diversity (diversity within samples), samples from the 2015 dataset had the highest
number of reads (pre-subsampling), OTUs, and Shannon and inverse-Simpson diversity amongst all datasets
(Table S3, Fig. S2). The 2015 samples also showed a wider range of alpha-diversity than the 2018 transition and
2017 late-melt samples, in line with the wider range of hydrological periods (and overall higher number of
samples) captured during the 2015 melt season (Fig. 2, Fig. S2). Late-melt 2017 samples exhibited lower alpha-
diversity than the 2015 samples, but were higher than the 2018 samples. Diversity in runoff samples generally
decreased over the sampling period in both 2015 and 2018, even if it was more pronounced for the 2015 dataset
(Fig. 2; Vrbická et al., 2022). However, whereas this deceasing trend was correlated with increasing Q in 2015,
no such trend was present in 2018. Instead, Shannon diversity in 2018 was positively correlated with SSC; for the
2015 samples a weak, though not significant (p-value > 0.05), correlation with SSC was also observed, but only
when excluding the early-melt samples (Fig. 2 d).

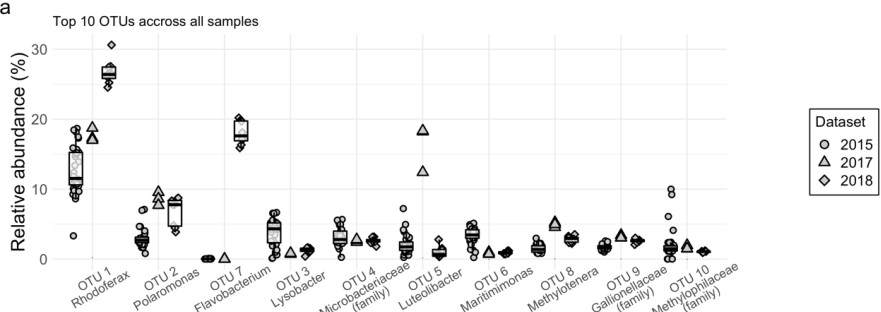

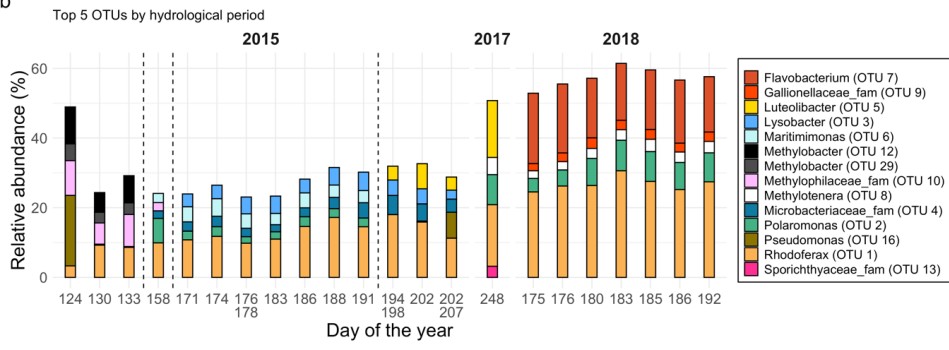


**Figure 3: Relative abundance of major OTUs.** (a) Relative abundance of the most abundant OTUs (top 10) across all
samples. Datasets are plotted separately with all replicate samples displayed. For 2015 and 2018 datasets, boxplots displays
median, interquartile range (IQR; box) and 1.5 X IQR (whiskers). On the x-axis, OTUs are ordered by relative abundance
across all samples, and OTU number and genus of the representative sequence displayed when possible; when the genus of



the OTU could not be classified, family is displayed instead. (b) Relative abundance of the most abundant (top 5) OTUs by
hydrological period. Average relative abundance of replicate samples are displayed. Vertical dashed lines in 2015 separate the
different hydrological periods. In 2015, a double day on the x-axis corresponds to samples that have been derived from two
separate days (see methods; Fig. 1, Table S2).

Looking at microbial assemblage composition, notable differences were also observed amongst the main OTUs
characterising the different hydrological periods. Whilst the most abundant (top ~2%) OTUs across all samples
were shared across all datasets (2015-18), there was a marked difference in their relative abundance between
datasets and hydrological periods. For example, the top 5 OTUs in 2017 and 2018 accounted for more than 50 %
of assemblage total relative abundance, while in 2015, they generally contributed up to ~30% (Fig. 3b). A marked
difference here were OTU 5 (*Luteolibacter*) and OTU 7 (*Flavobacterium*), which accounted for ~ 16-18 % of all
reads in 2017 and 2018 reads, respectively, but were nearly absent (< 0.01 % of dataset; OTU 7) or in much lower
relative abundance (1-2 % of dataset;  OTU 5) in the 2015/17 and 2015/18 datasets, respectively (Fig. 3). OTUs
related to *Methylobacter* (OTUs 12 and 29) were also overrepresented in the borehole and last set of 2015
samples (Fig. 3b).

Trends in beta-diversity (diversity between samples) better highlighted temporal changes in exported microbial
community structure over the different sampling periods. Different groupings between samples, hydrological
periods, and datasets are apparent on the PCoA and nMDS projections of Bray-Curtis dissimilarity (Fig. 4a, b*).
The most distinct communities were present in the 2015 borehole samples, but they were still more closely related
to 2015 runoff samples than those collected in late 2017 and 2018, with separations between datasets most notable
along the x-axis of ordination projections (Fig. 4a, b). Similar to alpha-diversity variations, the largest shifts in
communitiy structures were observed for the 2015 dataset, again likely reflecting the larger spread of hydrological
periods captured during the 2015 melt season (Fig. 1). A smaller, though relatively steady, temporal progression
was also observed for the 2018 samples (Fig. 4, b; Vrbická et al., 2022).

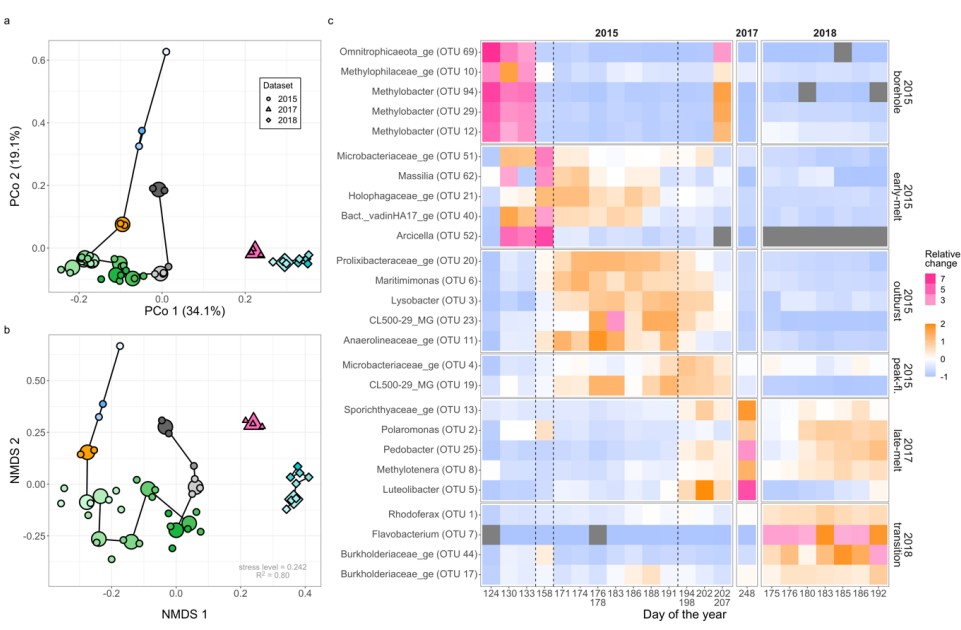




**Figure 4: Visualisation of microbial beta-diversity.** Ordinations of principle coordinates analysis (PCoA) (a) and non-metric dimensional scaling (nMDS) (b) on Bray-Curtis distances, and relative change in relative abundance of major OTUs idenfitied by Linear discriminant analysis effect size (LEfSe) (c). In a, b, shapes and colours correspond to the different hydrological periods and datasets respectively, as per Fig. 1-2. Smaller points represent individual replicate samples and larger points averages. Black lines link samples by sampling day for the 2015 and 2018 datasets (averaged points are linked when replicate samples are present). The colour gradient represents earlier (lighter) and later (darker) sampling days for each hydrological period. In (c), the heatmap displays relative changes in relative abundance accross the entire sample set. Two separate colour schemes are used to highlight trends for the more subtle changes in relative abundance (relative changes larger or smaller than 2); white indicates no relative change relative to entire sample-set average for that OTU, blues decreases in relative abundance relative to sample-set average and oranges and pinks increases in relative abundance relative to sample-set average. Dark grey tiles indicate absence of OTU in the sample. Only up to 5 OTUs per hydrological period (LEfSe class) are displayed (highest LDA score per LEfSE class; Table S4). Horizontal dashed lines separates hydrological periods on the x axis. Silva taxonomical classification of OTU representative sequences down to the genus level (when possible) displayed on the y axis (Table S5 for details).

We used LEfSe analysis to gain additional insight into the main microbial populations driving the differences in assemblage structure during the different hydrological periods identified. Looking at the relative change in relative abundance of these populations between all samples, we can see that specific OTUs were overrepresented during the different hydrological periods sampled (Fig. 4.c). LEfSE analysis reinforced trends highlighted in some of the most abundant OTUs identified above (Fig. 3), such as an overrepresentation of *Flavobacteria* (e.g. OTU 7) in the 2018 samples, *Luteolibacter* (OTU 5 ) in late 2017 and methylotrophic and methanotrophic clades (e.g. *Methylobacter*) in the 2015 borehole and the last set of 2015 samples (Fig. 4c). Additionnally LEfSe also highlighted a relative increase in OTUs related to specific ecological niches during the different hydrological periods sampled. For instance, a notable prevalence of OTUs related to (falcultative) anaerobes (e.g. Anaerolineales, OTU 11) or to sequences from anoxic systems (e.g. OTUs 20, 23) during the 2015 outburst period, versus OTUs related to taxa typically associated with more oxic environment (e.g. OTU 25), or those found on glacier's surfaces (e.g. OTUs 4, 7), overrepresented in samples from the 2015 peak-flow, 2017 late-melt, and 2018 transition periods (Fig. 4c; Table S5).

## 4 Discussion

The differences in microbial assemblages observed at LG between the different hydrological periods likely reflect contrasts in catchment water and sediment that principally supply the proglacial river at the time of sampling. The exact state of the glacier's hydrological system (i.e. water/solute sourcing and flowpaths) is difficult to infer and varies during and between seasons (e.g. when comparing 2015 and 2018 hydrological data during overlapping time periods; Fig. 1). Broadly speaking, water origin will develop from a higher proportion of basal and subglacially-stored melt earlier in the season, to a predominant contribution of (subglacially routed) surface meltwaters as the melt season progresses (e.g. Bartholomew et al., 2011; Chandler et al., 2013; Kellerman et al., 2020; Hatton et al., 2019). These variations in water sourcing and routing can explain the observed differences in microbial assemblage structure; these are discussed in more details below.

Early-season meltwaters more likely comprise a mix of groundwater, basal meltwater and stored surface melt from the previous season retained beneath the ice following drainage system shut down, that would have undergone biogeochemical transformation overwinter (Kellerman et al., 2020; Dubnick et al., 2023). These are



inferred to be stored in disconnected cavities, or water-saturated subglacial sediments closer to the ice margin
over-winter, and mobilised by new supply of meltwater from the ice sheet surface in spring (Chu, 2014).
More vigourous and episodic flushing of subglacial environments during outburst events mobilises more spatially
disparate sectors of the bed, but additionally likely mixes thicker layers of subglacial sediments encompassing a
broader range of environmental conditions (e.g. (micro)oxic-anoxic transition zones; Hodson et al. (2008)). In
2015, these outburst events were also accompanied by $CH_4$ pulses, indicating the flushing of methane-rich sectors
of the glacier's bed (i.e likely anoxic; Lamarche-Gagnon et al. (2019)). These geochemical patterns are consistent
with the increase in the relative abundance of putative anaerobic taxa exported to the ice margin during that period
(Fig. 4c; Table S4-S5), including methanogens (Fig. S3; see section below). As the glacier area experiencing
surface melting expands with snowline retreat, surface meltwaters enter the subglacial environment via moulins
at locations increasingly distant from the ice margin, and the evacuation of subglacial material from more remote
subglacial sources is indicated by the export of older particulate organic carbon (POC) within suspended
sediments during this period (Kohler et al., 2017). In short, the mobilisation of subglacial material from more
heterogeneous sources earlier in the melt season and during outburst events likely explains the overall higher
microbial diversity observed during those periods (Fig. 2a).
Similarly, a shift towards more streamlined and efficient flowpaths draining the glacier later in the melt season
likely explains the overall decrease in alpha-diversity with increasing Q in 2015 but also the relatively lower
diversity during late-melt in 2017 (Fig. 2). Following the 2015 outburst period, meltwaters are mostly flowing
through an already well-developed, hydrologically efficient drainage system where surface meltwaters rapidly
transit from the ice surface, to hydrologically efficient subglacial flowpaths, to the ice margin, and finally to the
proglacial zone (Chandler et al., 2013; Hatton et al., 2019; Chandler et al., 2021). This rapidly transiting surface
meltwater dilutes the contribution of subglacial waters with more prolonged contact with the sediments at the ice
sheet bed (e.g. slow and inefficiently routed and stored meltwaters). Towards the end of the ablation season (e.g.
2017 samples), surface melt input decreases but meltwaters continue to flow through an efficient subglacial system
previously developed earlier that summer that have not yet collapsed from ice overburden pressure (Davison et
al., 2019). Therefore, once the subglacial environment has been extensively flushed and drainage system
established and stabilised, we can expect a higher proportion of exported microbial assemblages to be derived
from glacial surfaces, as well as to consist of more stable and widespread (sub)glacial populations (e.g. putative
generalists; Dubnick et al., 2023). This is reflected here by a relative increase during these periods in OTUs related
to bacterial populations commonly found on glacial surfaces and/or reported in (sub)glacial systems more broadly
(e.g. OTUs 1 (*Rhodoferax*), 2 (*Polaromonas*), 4 (*Microbacteriaceae*), 5 (*Luteolibacter*), 25 (*Pedobacter*); Fig. 4,
Table S5; e.g. Zhang et al. (2012), Andrews et al. (2018), Kohler et al. (2020), Doyle and Christner (2022), Rathore
et al. (2022), Bradley et al. (2023), Dubnick et al. (2023)).
A very similar trend in beta-diversity progression linked to relative increased surface-related microbial
populations following melt-season progression has been reported in a another, smaller, GrIS catchment in South
Greenland (Dubnick et al., 2017). Sampling of a proglacial river on Disko Island (West Greenland) during late
August also revealed relatively high porportions of OTUs related to glacier surfaces in the microbial make-up of



the proglacial assemblages (Žárský et al., 2018). Overall, these results indicate that the composition of microbial
assemblages exported from the catchment is not only a function of discharge, but further highlights the importance
of timing, as microbial assemblages collected during low discharge early in the season differs from those collected
towards the end of the season during similar discharge (Fig. 1, 2, 4).

**4.1 Microbial data inform on the state of the LG hydrological system**

There was a stark difference in assemblage composition between the 2018 and 2015 samples which were collected
during the same time period. The most apparent microbal feature which sets the 2018 samples appart was the very
high relative abundance of populations from the order Flavobacteriales (respectively 21%, 4%, and 1% of all reads
from the 2018, 2015, and 2017 dataset; e.g. OTU7 Fig. 3). Nearest BLAST relatives of the most abundant
Flavobaceteriales OTU (OTU 7) are found in cryoconite hole sequences; i.e. glacial surface populations (Table
S5). Moreover, there was a lot of overlap in other OTUs overrepresented in the 2018 samples with those
highlited above for the 2015 peak-flow and 2017 late-melt samples (e.g. OTUs 1, 2, 4, 25); i.e. OTUs related to
glacial surfaces and/or widely reported glacial populations (Fig. 4, Table S5). The 2018 samples also exhibited
the lowest levels of alpha-diversity (Fig. 2). These microbial data suggest that at the time of sampling, a larger
component of microbial sequences were sourced from glacier surfaces and/or a smaller component from more
remote or heterogenously dispersed subglacial sediments in 2018 than at a similar time in 2015.
By taking a wider look at the overall 2018 melt season, it appears as if the 2018 sampling period occurred during
a slowdown or stasis in hydrological expansion for that sector of the ice sheet. The main melt onset (spring event)
in 2018 at LG seemed to have occurred around the 7 June 2018 (peak on the Watson hydrograph on day 158, Fig.
1), with discharge initially falling during the microbiological sampling period, and only increasing in intensity
toward the end of the sampling period (Fig. 1). This 2018 melt hiatus is likely explained by the relatively colder
air-temperatures in 2018 during the sampling period compared to previous seasons (Fig. S4). Of note is also the
lower pH recorded during the 2018 sampling period (pH ~ 6.5-7.5) compared to previous years at LG (pH ~ 7.2
– 9.8 over the same period; Fig. S5). Higher pH appears to be reflective of waters transiting through a subglacial
system where active physical erosion has produced an abundance of freshly comminuted reactive rock flour
conducive to rapid silicate hydrolysis and carbonation – this usually occurs during periods of increasing discharge
(Hatton et al., 2019). As such, the 2018 observation period is likely more analogous to a period following a
subglacial outburst or spring event, when surface waters flow subglacially through sectors of the bed with efficient
well-established subglacial drainage pathways that have been already flushed earlier in the melt season, although
closer to the ice-margin than  during peak flow following the outbusrt period.
Similar to the 2018 dataset, Flavobacteriales also represented a large percentage of LG runoff microbial sequences
collected during the 2012 melt season (May-September 2012; Cameron et al. (2017)). Given that 2012
corresponded to a historic record melt year in Greenland (Nghiem et al., 2012), and that more than half of the
Cameron et al. (2017) microbial dataset was collected following the 2012 outburst period and the record 2012
surface melt pulse, the high abundance of Flavobacteriales sequences reported there for LG might also reflect the
relatively large input of ice surface microbial populations in the LG runoff dataset at that time. Note here that the
relatively high surface contribution in 2018 (and 2012) is a relative interpretation comparative to other melt



seasons at the same site, and that the microbial make-up sampled at LG still bears an overall "subglacial imprint",
especially when compared to other glacial rivers in the region (see Vrbická et al. (2022)).
**4.2 Trends in methane-cycling populations at LG**
Both methanogenic and methanotrophic populations identified here have been reported previously in the context
of methane export from the LG catchment (Lamarche-Gagnon et al., 2019). A closer look at the temporal trends
in their relative abundance sheds some light onto their distribution beneath the ice. The relative abundance of
methanotrophic populations was highest in borehole samples beneath the river ice in front of LG (~10-20% of
reads; Fig S3). This is consistent with the relatively high methane concentrations, reduced $O_2$ availability and
stable conditions *in situ* (Table S1). How widespread these conditions are beneath the ice sheet, however, is
unclear, but very similar methane-oxidizing populations have been reported for example in the (micro)oxic and
methane-rich water and sediment layers of subglacial lakes in Antarctica (Davis et al., 2023; Achberger et al.,
2016)*. Near identical methanotrophic populations were also overrepresented in small methane-rich subglacial
outflows of the same GrIS sector throughout the 2012, 2018 and 2019 melt seasons (Dieser et al., 2014; Znamínko
et al., 2023). Unlike for the LG river here, however, these relatively smaller outflows drain much smaller areas of
the bed constrained near the ice margin and might not be representative of ice sheet environments at large
(Hawkings et al., 2021b).

In 2015, the relative abundance of methanotrophs sharply decreased during the early-melt and outburst periods at
LG (Fig. S3, 4c), when subglacial waters from more remote sectors of the glacier bed were being progressively
accessed. An increase in relative abundance then occurred in the last set of 2015 samples (26 July) when waters
were rapidly transiting from the glacier's surface to the margin in well-established subglacial conduits (*see above
section*). A very similar trend in relative abundance between aerobic methanotrophic bacteria and putative
anaerobic methane-oxidising archaea was observed in 2015 (Fig. S3), suggesting that if anaerobic methane
oxidisers are present beneath the ice, they might occupy a similar niche to aerobic methanotrophic populations,
perhaps relying on oxidised iron (which is likely in adundance; Hawkings et al. (2014)) as terminal electron
acceptor (Cai et al., 2018). Almost inversely, an increase in methanogen relative abundance was observed during
the 2015 othe early-melt and outburst periods, which aligns with the methane pulses observed during these periods
(Lamarche-Gagnon et al., 2019). In 2018, methanotrophic bacteria accounted for a larger proportion of
assemblages than in most of the 2015 runoff samples (Fig. S3). As discussed above, waters sampled in 2018 were
likely rapidly transiting in established efficient subglacial flowpaths, likely closer to the ice-margin than at a
similar period in 2015. Together, these temporal changes in relative abundance suggest that subglacial
methanotrophic niches may be constrained to the ice margin or (flanking) efficient drainage flowpaths in that
sector of the GrIS, with methanogenic hotspots in deeper sediments beneath the ice-rock interface.
**5 Conclusion**
The trends and differences in microbial composition during and between melt seasons observed here highlight
that the timing of sampling can influence the conclusions one can derive from spot sampling, or even continuous
sampling during a short period of a melt season without a more complete overview of seasonal hydrology and



hydrochemistry. We show that sampling the microbial assemblage from the same site thoughout a melt season
and in multiple years can yield different results, primarily because of changing hydrological regime. Therefore,
depending on the information details desired during microbial investigation of these dynamics systems, a good
understanding of glacial hydrology and hydrochemistry might be critical in informing microbial data. This is
likely especially true in larger glacial catchments that also undergo more dramatic hydrological re-configuration
over and between melt seasons.
**6 Code availability**
mothur .logfiles and .batch files will be uploaded to the Dataverse repository associated to this manuscript prior
to publication (https://dataverse.no/dataverse/uit).
**7 Data availability**
**7.1 Molecular data**
Raw reads for the 2015 dataset are available from the NCBI Sequence Read Archive
(https://www.ncbi.nlm.nih.gov/sra) under BioProject PRJNA495593. Quality checked datasets for 2017 and 2018
have been deposited in the MG-RAST database (https://www.mg-rast.org/) under the accession number
MGP92375 and MGP104407 respectively.
**7.2 Hydrological and hydrochemical data**
Hydrological sensor data for the LG river in 2015 and 2018  (Q, SSC, pH,  as well as electrical conductivity), can
be found at: https://zenodo.org/doi/10.5281/zenodo.4634279 (Hawkings et al., 2021a). Hydrochemical data for
the 2015 melt season can be found here: https://doi.pangaea.de/10.1594/PANGAEA.896788 (Hatton et al., 2018).
The discharge record from the Watson river can be found here: https://doi.org/10.22008/FK2/XEHYCM (Van As,
2022)  and  the  automatic  weather  station  PROMICE  data  can  be  found  here:
https://doi.org/10.22008/FK2/IW73UU (How et al., 2022).

Hydrochemical data from the 2015 borehole water samples, as well as manual pH measurements from the LG
river for the years 2009, 2010, 2012 will be uploaded to the Dataverse repository associated to this manuscript
prior to publication (https://dataverse.no/dataverse/uit).
**8 Author contribution**
GLG, MS, AMBA, JH and JLW conceived the project. GLG, MS, JLW, JH, PK, JK, TJK, LB, JH, AB and JT
collected the samples. GLG, MS, PK, JK, TJK and JH performed the lab work. GLG performed the analyses with



LF and KV contributing to data curation. GLG wrote the manuscript along with significant input and editing from
all coauthors. All authors contributed to the article and approved the submitted version.

**9 Competing interests**

One of the (co-)authors is a member of the editorial board of TC.

**10 Acknowledgments**

The authors acknowledge past and present Greenlandic people as stewards of the land where this research took
place. Greenland terrestrial research campaigns were funded by a UK NERC standard grant (NE/I008845/1) and
a Leverhulme Trust Research Grant (RPG-2016-439) to J.L.W., with additional support provided by a Royal
Society Wolfson Merit Award to J.L.W. and from Czech Science Foundation grants (GACR; 15-17346Y and 18-
12630S) to M.S. DNA analyses and sequencing were funded through a Czech Ministry of Education, Youth, and
Sport grant (ERC CZ LL2004) to M.S. and a UK NERC grant (NE/J02399X/1) to A.M.A. This research was also
part of a European Commission Horizon 2020 Marie Skłodowska-Curie Actions fellowship ICICLES (grant
agreement #793962) to J.R.H.
We also extend our thanks to all those involved with fieldwork at Leverett Camp during the 2015 and 2018 field
campaigns, particularly Marie Bulínová and Stefan Hofer, as well as the Kangerlussuaq International Science
Station (R. Møller and C. Lager) for support with field logistics. A special acknowledgement also to Pat Schloss
and the Riffomonas Project and YouTube channel (riffomonas.org; https://www.youtube.com/@Riffomonas) for
their freely available and extensive online resources on microbial and environmental data analysis and
visualisation.

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
