# Peer review of "Understanding microbial sourcing in Greenland subglacial runoff"

_EGUsphere, 2024_

## Author Comment (AC1)

**Response to reviewer 1**

We thank the reviewer for their extensive review. We first try to address the general comment on potential methodological artefacts below and then response to specific comments individually. Our response to comments from reviewer 2 is also included following responses to reviewer 1. Reviewer comments are italicised (sans-serif) and our responses not (serif font). Specific comments are also numbered for reference.

Response to general comments

*[…] I have my reservations about several of the phylotypes/OTUs identified, most notably those identified in 2018 (e.g., Flavobacteriales) that were not identified in 2017 or 2015. Likewise, no Lysobacter in 2018. This would factor in largely to the patterns observed yet it is also possibly artifact due to the use of other primer pairs in 2018 than in prior years, the use of additional purification steps in 2015, and the use of different sequencing centers to generate these data. Other OTUs (e.g., Rhodoferax, Polaromonas) perhaps behave more predictably (slight increase then decrease during hydro cycle) and are more cosmopolitan across years. I suggest that the authors remove OTUs that are not present across years and reperform their analyses to see if they still see the patterns emerge at a compositional level.*

We agree that our data interpretations hinge on the different microbial datasets collected during different melt-seasons to be comparable and not significantly impacted by methodological artefacts, and we understand the concerns raised by the reviewer. However, we also feel that some of the data might have been misinterpreted by the reviewer, mainly Fig. 3b which only highlights the top 5 OTUs for each hydrological periods unlike in panel 3a where top OTUs are displayed for all samples.

For example, *Flavobacteriales*, as well as *Lysobacter*, OTUs were present and abundant in all datasets and from all collection years. The most abundant Lysobacter OTU in all samples (OTU3) was more abundant in 2015 (~ 4 % relative abundance), but was still present at relatively high relative abundances in the 2017 and 2018 datasets (~1%), as shown on Fig 3a and explicitly stated on Ln 453-454 of the initial manuscript (now modified according to comment on Cyanobacteria below). Regarding concerns raised towards the order *Flavobacteriales*, we believe the reviewer refers to the very high relative abundance in 2018 of the *Flavobacterium* OTU 7 (averaging ~ 18% relative abundance in 2018; Fig. 3a-b). However, here again this OTU is not absent from the other datasets/melt-years but present at a much lower relative abundance (<1%); we concede that this is not clear from Fig. 3a where it appears that OTU 7 relative abundance for 2015-17 might be 0 and will amend the figure legend to clarify this point. The significantly larger relative abundance of OTU7 in 2018 is discussed in more details in the manuscript and we agree with the reviewer that our interpretations need to be based on genuine environmental differences. We do recognise that primer biases may account for some minor differences between the different datasets (Ln 173-74 of the initial manuscript), but we struggle to explain how these might be responsible for such specific and significant differences. For example, we double-checked how well (or not) the target regions of the 16S RNA DNA sequence of OTU7 align against both primer pairs and they sit perfectly for both primer pairs used. In other words, we see no reason why or how the more degenerated primer pair used on the 2018 samples (Parada et al. (2016) 515F-Y and Apprill et al. (2015) 806R) would lead to preferential amplification of these DNA reads; if anything the opposite would be expected due to the higher level of degeneration and impact on PCR amplification. More importantly, both primer pairs detected several *Flavobacteriales* OTUs in all datasets, up to about 1 % and 4 % relative abundance in the 2017 and 2015 datasets respectively (data not shown). In the case of the second most abundant *Flavobacteriales* OTU, OTU 6 (*Maritimimonas*) − also

belonging to the same *Flavobacteriaceae* family as OTU 7 – it has a higher relative abundance in 2015 than in 2018 (~4% vs 1%, Fig. 3a). Moreover, the Caporaso 2011 primer pair used to generate the 2015-17 datasets is known to accurately target and amplify bacterial sequences belonging to the genus *Flavobacterium*; e.g. see the meta microbiome study on cryospheric ecosystems (Bourquin et al., 2022) that shows *Flavobacterium* genera to be core members of these microbiomes using the same primer pair used here for the 2015-17 datasets.

We consider that re-running the entire analysis with only shared OTUs unnecessarily time consuming and also slightly artificial, and do not see how this would largely impact our results and interpretations as they would mostly (only) relate to very rare taxa. As a demonstration exercise, we re-ran our beta-diversity ordination analysis on a dataset excluding the *Flavobacterium* OTU 7. As pointed out above, OTU 7 is present in all datasets but accounts for a significant fraction of the 2018 community's relative abundance. Therefore, removing this abundant OTU would somewhat impact beta-diversity analyses. As seen in the ordinations below, removing OTU 7 from the analyses still sets the 2018 samples apart from those collected in 2015, but has for main effect to cluster them more closely to the late-melt 2017 samples. If anything, this reinforces the messages and discussions of the manuscript highlighting the similarities between both hydrological periods relating to glacial hydrological conditions at the time despite very different times of the year (see 4.1), where and when glacial waters likely access and flush out similar environments and associated microbial communities.

[Figure]

For all of the reasons stated above, we maintain that the differences in OTU relative abundances between hydrological periods and datasets are mostly caused by environmental factors and not by methodological artefacts, as discussed in the manuscript.

As stated above, we feel that perhaps the reviewer misinterpreted the datasets due to the panel "b" of Figure 3, where only the top 5 most abundant OTUs by hydrological periods are displayed to avoid overplotting. That is, the absence of an OTU-colour bar in one hydrological period compared to another doesn't imply that such an OTU is absent from samples in another hydrological period, but rather that such an OTU doesn't fall within the top 5 most abundant for a given period. E.g. OTU 10 on Figure 3b (pink bar) is only visible on the 2015 borehole and early-melt stacked bar plots, but it can be seen on Fig. 3a that OTU10 is present in all samples (of on Fig. 4c). Same thing applies for OTU 7, 9, 8 etc. In fact, only the most abundant OTU across all samples/datasets, OTU1 (*Rhodoferax*), is visible on all stacked bar plots on Fig. 3b. Again, we decided to limit plotting 5 OTUs per hydrological period on Fig. 3b to emphasise variabilities amongst the most abundant OTUs between the different hydrological periods. However, we do see now how Fig. 3b might be misinterpreted and we will modify the figure legend to clarify the points above – the legend will now read:

*"[…] (b) Relative abundance of the most abundant (top 5) OTUs by hydrological period. Average relative abundance of replicate samples are displayed. Vertical dashed lines in 2015 separate the different hydrological periods.* **Note that only the top 5 most abundant OTUs by hydrological periods are displayed to avoid overplotting and that the absence of an OTU in one hydrological period on the figure does not signify its absence from corresponding samples (e.g. compare with Fig 3a).** *[…]"*

*Further, based on the methods and locations of sampling, I don't think that a definitive statement that the same site was sampled throughout the melt season can be claimed (line 522).*

We maintain that the slightly different sampling locations (< 2 km apart) during and between collection years would not significantly impact the observed microbial community composition (assemblages) given the absence of significant tributaries within that reach, the order of magnitudes larger subglacial vs pro glacial catchment area upstream of the sampling locations, as well as the relatively fast flowing waters (~1 m s-1) and substantial discharge (often >100s m3 s-1; Fig. 1) of the LG river which would not allow for any meaningful ecological processes to occur within the first 2 km of river sampled, as stated on Ln 151-157 of the initial manuscript.

We would like to re-stress the fact that, overall, we do not consider the differences in microbial composition observed to be significantly impacted by *in situ* processes and changes occurring on the time scale of overall riverine transport upstream of the sampling site (hours to < 1 day – e.g. Chandler 2013), but instead reflect differences in sourcing environments (e.g. degree of mixing between supra- vs sub- glacial environments, (a)noxic subglacial environments, subglacial proximity to the ice margin, more or less access to environments impacted by overwintering or longer subglacial residence time etc.) due to differences in glacial hydrological drainage and hydrological connectivity during the different hydrological periods. We have tried to re-emphasise and clarify this thesis/narrative in the introduction/discussion based on further comments and responses below (e.g. see responses to comments 11-13).

Response to more specific comments

1.  *Lines 16-18: How can one make this statement if these systems are so poorly understood. My recommendation - delete the first part of the sentence or reword to indicate that their contribution to biogeochemistry is poorly understood due to logistical challenges.*

    Will re-phrase and also include additional references as per comment to reviewer 2 below. Will now read:

    *"The beds of glaciers and ice sheets are now recognised as widespread ecosystems conducive to a range of biochemical and biogeochemical reactions in situ (e.g. Sharp et al., 1999; Hodson et al., 2008; Hamilton et al., 2013; Christner et al., 2014; Davis et al., 2023). The global significance of these active subglacial ecosystems is still unclear but mounting evidence suggest far-reaching impacts extending well beyond beyond the ice margin (Wadham et al., 2019)."*

2.  *Lines 19-20: I don't disagree but as far as I can tell, this is the hypothesis to drive the work rather than it being definitively known.*

    Here we are referring to the origin of "exported material" in an all-encompassing term, including all particulate (e.g. sediments) and dissolved material, not only microbial cells. We believe the fact that geographically separated regions of the ice-sheet bed (that might comprise heterogeneous material) are being accessing during different periods of the melt-season to be widely recognised and accepted. E.g. It has been demonstrated that the particulate organic carbon of suspended sediments exported from LG varies in 14C over the melt-season, indicative of mixed or different sources (Kohler 2017; also referenced in the text at multiple occasions). Similarly, the degree of surface vs sub-surface contribution to proglacial runoff (stored waters, ions, etc.) varies over a melt-season. We therefore believe this statement to be factual and not the hypothesis of the manuscript. Part of our hypothesis relates to whether or not, and to what extent, this heterogeneity is also reflected in the microbial assemblages of the meltwater river and whether or not it can inform on microbial communities prior to their entrainment by meltwaters.

3.  *Line 77 and throughout: I imagine geochemists/hydrologists will read this and wonder what is meant by assemblages in this context. Please refer to assemblages more descriptively (e.g., microbial assemblages or communities)*

    Good point and will clarify throughout.

4.  *Lines 158-159: Given the effect of minerals on DNA recovery and purity, and downstream effects on PCR, i cannot imagine that this additional purification did not effect the outcomes. Can the authors demonstrate that this has an effect (or not) using an existing sample, one of which is purified like this versus not and then sequenced to determine the introduced bias. I also wonder why replicate subsamples were not sequenced but were rather pooled, at least for one sampling point to assess variation that may exist?*

    We acknowledge that comparing datasets produced using slightly different methodologies is not ideal. Unfortunately, we cannot demonstrate using existing samples direct sequencing results from purified vs non-purified DNA samples. We understand the concern regarding the

effect of mineral and glacial flour on DNA recovery and, indeed, DNA concentrations were generally very low for samples containing high sediments loads. Even though not informative regarding sample purity, it is worth noting that DNA concentrations and yields were comparable regardless of the DNA purification step in all samples (usually below 0.6 ng/µL). We would like to emphasise, however, that this paper aims to compare, as best as possible, datasets for the same study site but that have been produced for different studies (i.e. Lamarche-Gagnon et al. 2019, Kohler et al. 2020 and Vrbická et al. 2022) and therefore that absolute methodological replicability was impossible. We still believe that the minor methods differences between the different datasets do not significantly impact results and interpretations. Furthermore, can see for example Sáenz et al. (2019) regarding the minor to no impact of DNA purification steps on Illumina MiSeq 16SrRNA libraries from environmental samples.

Regarding the pooling of some replicate samples for the 2015 dataset, it was performed in an effort to increase DNA concentrations prior to sequencing given the low DNA recoveries mentioned above. This step was only performed for some samples and, as detailed in Table S2, several replicate samples were sequenced without a pooling step (e.g. L174_15, L183_15, L191_15, etc) and no obvious differences in microbial community variation/variability is present for the pooled versus non-pooled sample sets (e.g. beta-diversity as observed on Fig. 4a, b).

5. *Lines 190-191: These would seem to be possibly informative since they too could point to input of surface materials and photosynthetic biomass/organisms.*

We agree that good data on chloroplasts and Eukaryota (e.g. glacial algae and fungi) could provide useful information, especially regarding the contribution of glacial surface populations to the overall microbial communities and assemblages. However, the universal primer pairs used in our study is designed and optimised for bacterial and archaeal sequences and we decided to follow standard bioinformatics procedures that remove reads classified as mitochondria or chloroplasts which are normally associated to unspecific amplification. Re-adding chloroplast sequences would require an entire re-run of all analyses. Again, we agree that such information could be valuable and future studies might want to include those in their analyses if they can be accurately amplified and classified.

6. *Line 209: I don't understand this. Perhaps inserted due to a prior reviewer comment? Regardless, the authors did about as much with their putative OTUs involved in CH4 cycling than they could have.*

We will remove this sentence from the manuscript.

7. *Lines 216-218: 'Candidatus Methanoperdens nitroreducens' and then 'Cand. M. nitroreducens' thereafter.*

Thank you. We were refering to *Candidatus Methanoperedens ferrireducens* (which is very much related to *Cand. M. nitroreducens*) but indeed the species name got lost in this version of the

manuscript. Will correct to "*Candidatus Methanoperedens ferrireducens*" and then "*Cand. M. ferrireducens*" thereafter.

8. *Figure 2c and 2d. Does this then suggest that the drivers of community composition are different depending on the time by which you sample them? In other words, is this stochastic (mixing of different source waters) rather than deterministic (true environmental drivers) that are difficult to deconvolute? This seems to be likely as i cannot imagine that what the authors are seeing is community turnover but rather is the former (mixing of different communities that themselves are deterministically structured).*

In short, yes. We interpret the variability in microbial community composition to variabilities in water and sediment sources due to changes in glacial hydrology, and not as pointed out by the reviewer here to population turnovers of a same overall glacial microbial community. This is why the title and abstract of the study mentions "microbial sourcing" or "source environment" of the microbial populations and communities making-up the microbial assemblages observed in the proglacial river. We wanted to make this point clear in the introduction (Ln 49-50 and 56-65). To what extent one can deconvolute or not the mixing (stochasticity) of the different water and sediment sources hosting different communities (themselves structured by environmental drivers in situ, or "determinism") makes the core of the discussion. Regarding our interpretations of Figure 2c-d more specifically, we believe these are explained Ln 420-435 of the initial manuscript.

To better reflect our position, we will modify the last sentence of the first paragraph of the discussion:

*"[…] We believe that these variations in water sourcing and routing, **rather than en route community turnovers responding to changes in environmental forcings**, better explain the differences in microbial assemblage structure observed in the proglacial river throughout and between melt-seasons (e.g. Dubnick et al., 2017); these are discussed in more details below."*

9. *Figure 2 and more generally: Can the authors estimate biomass loads from their DNA recoveries and the volumes of material recovered. It would be interesting to see how this might have changed over a seasonal melt cycle.*

Unfortunately, we are not sure how we could estimate microbial biomass based on our DNA recoveries. A previous study slightly downstream from our study site (cited in the manuscript), however, has measured relatively stable cell concentrations in Greenland proglacial runoff on the order of $10^{4-5}$ cells/mL over the 2012 melt-season (Cameron et al., 2017). Interestingly similar (perhaps slightly reduced) order of magnitude ($10^4$ cells/mL) generally found in supraglacial runoff (Stevens et al., 2022; Andrews et al., 2018).

10. *Lines 319-321: But the read number is irrelevant correct, since the sequence library was downsampled prior to calculation of alpha diversity, correct?*

The read number was for reference only and corresponds to read numbers prior to "downsampling" (i.e. pre-subsampling), as indicated on Ln320 of the initial manuscript. The

reviewer is correct in that sequence libraries were downsampled prior to downstream diversity analyses and that little (if anything) can be derived from this information (again, mentioned here for reference only).

11. *Lines 400-401: This could be tempered based on above comments.*

In line with and to better reflect our response to the previous and below comments on stochasticity and determinism (comment 13), we will re-phrase to:

*"We believe that these variations in water sourcing and routing, better explain the differences in microbial assemblage structure observed in the proglacial river throughout and between melt-seasons (e.g. Dubnick et al., 2017), rather than en route community turnovers responding to changes in environmental forcings. Placed within ecological theosry, this suggests that microbial assemblages in proglacial runoff are mostly driven by stochastic processes (dispersion) rather than determisnitic ones (ecological selection)."*

12. *Lines 406-407: I would imagine these would also be the most anoxic of the sample regimes and the community composition would reflect this. Yet, the communities are dominated by putative aerobic methanotrophs...*

Here we were mostly referring to early-melt waters prior to the onset of the outburst period (e.g. L158_15 samples) which were not dominated by putative aerobic methanotrophs but instead did show a relative increase in putative anaerobic clades compared to borehole, peak-flow, late-season, and 2018 samples (Fig. 4b, Fig.S3, Table S5); we believe the reviewer is instead referring to the pre-melt river-ice borehole samples (L124-133 2015) which indeed were dominated by putative aerobic methanotrophs (Fig. 4c, Fig. S3a) and were not completely anoxic, despite themselves almost certainly representing also a mixture of groundwater, basal meltwaters, and late-melt from the previous seasons (Ln 246-253 of initial manuscript). Given the borehole samples consisted of the earliest set of samples presented, we understand the confusion on Ln406-07.

We will modify the text to clarify which samples are discussed here, and that borehole water samples are also discussed later in section 4.3. The second paragraph of the discussion will now explicitly mention borehole samples and how these differ from runoff waters and also allude to the now previous statement on stochasticity and determinism:

*"A nuance to the above statement relates to the earliest set of samples presented here (days of year 124-133 2015) collected prior to melt onset in 2015 beneath the frozen river (borehole samples) and which did not consist of flowing meltwaters. Unlike the rest of the described samples from proglacial runoff, the microbial assemblages from these borehole waters likely better reflect measured environmental conditions in situ (Table S1) given the lack of significant influence by mixing and reorganisation caused by rapidly flowing meltwaters. The methane-cycling communities present in those borehole samples are further described in section 4.2 below."*

We also feel that early-melt water samples and outburst samples can be discussed simultaneously and will combine those section into a single paragraph (Ln 403-421 of the initial manuscript).

13. *Lines 419-421: A good place to include discussion of mixing and stochasticity and determinism and their relative importance in explaining what is observed.*

See our response to comment 11 above, which we feel also address the point raised here.

14. *Line 460: Again, I would be looking for chloroplast sequences. Were Cyanobacterial sequences identified at all that could help?*

We agree that Cyanobacteria sequences, like chloroplasts, could be a potential marker for surface communities and could have been a more elegant target to discuss contributions from surface populations to proglacial runoff. However, Cyanobacteria sequences were only present in very low relative abundance (<0.1%) in all samples and as such we do not consider them to provide significant insights for the discussion and why we instead focused on other more abundant OTUs. We do realise that Cyanobacteria are normally associated to glacier surfaces, for example in cryoconite holes, and that their very low relative abundance in our datasets may at first seem to counter some of our discussion in section 4.1. However, Cyanobacteria do not necessarily make-up a significant fraction of cryoconite hole communities.

In fact, Flavobacteria (e.g. OTU 7) are also very common on glacier surfaces (including in Southwestern Greenland) and can even dominate (>80%) cryoconite holes communities (e.g. Zawierucha et al., 2022; Zhang et al., 2006; Zhu et al., 2003; Xiang et al., 2009; Foreman et al., 2007; Poniecka et al., 2020). Given the exact match (100% identity) between OTU 7 representative sequence and Greenland cryoconite Flavobacteria (Table S5), we are confident that these identified populations likely originated from the ice-sheet surface as discussed.

However, we now realise that probably more references to the above literature on cryoconite Flavobacteria populations should be included, as well as mentioning the low relative abundance of cyanobacteria. We have amended section 4.1 accordingly.

15. *Lines 492: I don't understand this - the reduced O2 availability would likely select against methanotrophs and rather favor methanogens. Are the authors suggesting that the low O2 is due to consumption? if so, then why not complete consumption of methane*

Was mean to highlight the presence of O2 availability for methanotrophy but at sub air-saturation concentrations. We acknowledge the confusing wording and will re-phrase to:

*"This is consistent with the relatively high methane concentrations, the presence of dissolved $O_2$ although at hypoxic levels, and stable conditions in situ (Table S1)."*

16. *Lines 508: Anaerobic methanotrophs and aerobic methanotrophs don't share a niche. Their niche might partially overlap in that they both oxidize CH4 but O2 will partition them beyond that. Suggest modifying this statement to avoid confusion.*

Will modify to:

*"A very similar trend in relative abundance between aerobic methanotrophic bacteria and putative anaerobic methane-oxidising archaea was observed in 2015 (Fig. S3), suggesting that if anaerobic methane oxidisers are present beneath the ice, their niche distributions might partially overlap those of aerobic methanotrophic populations, perhaps relying on oxidised iron (which is likely in abundance; Hawkings et al. (2014)) as terminal electron acceptor (Cai et al., 2018)."*

**Response to reviewer 2**

Response to general comments

*As a microbial analysis of several years of Leverett Glacier samples, "Understanding microbial sourcing in Greenland subglacial runoff" provides an excellent look into the variability of microbial communities in emergent Greenland Ice Sheet waters. The combined data from multiple years and multiple hydrological regimes shows a far more complex picture than a single shorter study could. I have several suggestions for improving the manuscript that should be possible to address in the course of revisions. In general, I think the introduction needs work and several points in the discussion are not necessarily well supported. The supplemental tables also need work.*

We thank the reviewer for their positive and extensive review. We hope that our responses and modifications address their concerns.

Response to specific comments

1. *Line 38: Consider a word other than "leak", which implies built infrastructure. "Evade" perhaps?*

We will replace to "escape"

2. *Lines 41-50: I don't think marginal sampling should be presented as some sort of work around. The glaciohydrological system naturally samples the subglacial environment, and I don't think anybody has ever seriously suggested that what emerges from a glacier differs importantly from what's beneath it. The very first work that demonstrated widespread bacteria in the glacial bed (Sharp et al., 1999) combined samples collected in boreholes with marginal samples, and did not find an appreciable difference in the bacterial content. There are a number of studies that have directly accessed the subglacial environment in Antarctica (oddly are not cited here). But in Antarctica, marginal sampling of subglacial waters is generally not possible, as the vast majority of subglacial waters discharge into the ocean under ice shelves. Thus, Antarctic biogeochemistry is largely limited to drilling projects; whereas in Greenland and alpine settings, nature can do the sampling for us. Not to say drilling projects aren't useful or interesting in these settings. Graly et al. (2014) find some differences in water chemistry between boreholes and the integrated outwash at Isunnguata Sermia, and similar things have been observed in alpine glaciers (e.g. Tranter et al.). But all of these differences have glaciohydrologic explanations (i.e. boreholes being more isolated, etc.). I think this section should be reframed and probably expanded to a broader selection of literature.*

We recognise and fully appreciate the advantages and the opportunities that marginal sampling in Greenland (and other alpine or valley glaciers) raised by the reviewer can offer, as well as the underlying glacial hydrological mechanisms controlling subglacial runoff. In fact, the entire manuscript relies on this understanding of glacial hydrology to explain the variabilities in (or lack of) and composition of microbial assemblages present in glacial runoff collected via marginal sampling. We therefore don't really see how the paragraph on Ln 41-50 currently disagrees with the reviewer's comments given that it does acknowledge that marginal sampling does offer alternative to drilling. In the context of microbial ecology, however, we maintain that

the use of marginal sampling has limitations to fully understand (sub)glacial microbial communities, essentially due to the integrative nature of meltwater runoff in terms of the microbial habitats it combines and mixes *en route* to the ice margin, and which should vary over and between melt-seasons (the premise of this study). We believe that both advantages as well as limitations of marginal sampling for microbial ecology studies are described in Ln 41-50.

We will however slightly nuance the statement on Ln 40 to: "*Despite this importance, our understanding of subglacial microbial processes and communities remains limited, partly hindered by the difficulty to directly access the subglacial environment.*"

We also appreciate and understand the glacial hydrology differences between Greenland and Antarctica but we recognise such differences had not been explicitly stated in the manuscript. We will modify Ln 44 of the initial manuscript to clarify this: "*In most glacial systems outside of Antarctica, such as those found in the ablation zone of the Greenland Ice Sheet (GrIS) […]*"

Regarding the lack of Antarctic literature cited in the introduction, the very first sentence is referring to Antarctic subglacial lake research (Christner et al. 2014). We will now also add the reference Davis et al. (2023) on the Antarctic Mercer Subglacial Lake, which currently is only referenced later in the manuscript (Ln 495). We do thank the reviewer for highlighting the absence of the seminal study by Sharp et al. 1999, and completely agree that it should be included in the introduction and it will be added to the first sentence. See our response to comment 1 of reviewer 1 above.

3. *Lines 52-65: This paragraph should probably cite more literature. There were quite a few insights into glaciohydrology developed between the theoretical work of the early 1970s and Davidson et al. (2019). There's a certain facile element to the framing/presentation here. The notion that temperate glaciers change seasonally is not exactly novel. Maybe the references cited in lines 87-89 could be moved forward to this paragraph, but with an actual explication of their findings.*

We indeed appreciate the extent of glacial hydrology studies of the past decades and did not intend to imply the opposite. The choice to reference Davidson et al. 2019, an extensive review paper that details these hydrological complexities and nuances which we believe go well beyond the scope of the present study, in order to reduce the number of citations here, especially given that more references are cited in the methods section, as pointed out by the reviewer. We see, however, how only citing foundation studies on alpine glaciers from the 70s and a review on Greenland hydrology in 2019 can seem restrictive and misleading. We will amend and only cite Davison et al. 2019 (which does review and reference these earlier studies) but with the caveat

"(e.g. *see Davison et al. 2019 and references therein for an extensive review*)".

Again, the aim of this study is to make use of the well-established glacial hydrological science and try to better apply it to microbial ecology studies, and potentially vice-versa. To clarify these points and in response to comments from both reviewers (including comment 3 here), we will modify the paragraph and reference list. E.g. amendment to this paragraph will now include:

*"Even if surface melt always make for the bulk of the water present in proglacial runoff, the degree of mixing between surface melt and the subglacial environment subsequently varies, influencing the quantity and the origin of (sub)glacial material that is transported to the ice margin (e.g. Linhoff et al., 2017)."*

Also see response to comment 4 below which we believe addresses some the points raised here.

4. *Line 64: Be more precise than simply saying "older". In a geologic sense, the bed material is billions of years old at this site. This is discussing the sediment residence time, i.e. the time between when subglacial materials enter the glaciohydrologic system and when they leave it.*

Older here was meant to reflect sediment organic matter bearing a more depleted 14C signature (e.g. Kohler et al. 2017). This will be rephrased as:

*"This is especially true for samples from terminal outlets of larger glacial systems, which undergo more dramatic, pronounced and complex hydrological change throughout the melt season, draining more expansive areas as meltwater generation increases, and likely exporting to the proglacial zone bed material that has been isolated beneath the ice sheet for longer periods of time (e.g. older organic carbon; Kohler et al., 2017)."*

And followed by:

*"Overall, microbial assemblages present in meltwaters should reflect their source environments and respective degree of mixing upstream of the sampling site. These will change over and between melt-seasons, which can make disentangling the origin of exported material challenging if no additional information on the state of the glacier's hydrological system is considered (e.g. Dubnick et al., 2017)."*

5. *Line 68: I would avoid using the abbreviation LG. Non-standard abbreviations have the potential to confuse readers. It is not needed for space reasons. I would also avoid GrIS. OTU might be okay, simply because it is used so frequently. But LDA, LEfSE, PCoA, nMDS, etc. become a garble to those not deeply familiar with these methods.*

LG is used 41 times throughout the manuscript and therefore believe that not using the abbreviation will significantly burden the text. Similarly, GrIS used 9 times and we believe this abbreviation to be very familiar to the audience of The Cryosphere journal. Regarding methodological abbreviations, those are commonly used in microbial ecology and staples of 16S rRNA studies and defined in the methods section. We have, however, included the entire definition of the "LEfSe" abbreviation in corresponding methods title as this was missing.

6. *Line 76: I would use glaciofluvial rather than fluvial here. The uninformed reader might think you are referring to a normal subaerial river.*

Changed

7. *Lines 80-84: The methane question should be more properly developed, if this is a major aim of the paper. I.e. disentangle this list of references to state what's known and unknown about methane cycling in this sector of the Greenland Ice Sheet.*

These references all allude to either the potential for subglacial sediments to host active methanogens (Stibal 2012) and/or report on subglacial emissions bearing a microbial methane signature (rest of the references). We will disentangle and clarify the messages from the list of references as follow:

*"Given the increased attention surrounding reports of subglacial emissions of microbial methane from this sector of the ice-sheet  (e.g. Dieser et al., 2014; Christiansen and Jørgensen, 2018; Lamarche-Gagnon et al., 2019;*

*Pain et al., 2019), as well as the impacts of bacterial methane-oxidation on mitigating glacial emissions (Dieser et al., 2014; Michaud et al., 2017; Žnamínko et al., 2023), we further apply our interpretations to focus on putative methane-cycling populations to see what additional information can be gained about methane cycling beneath the GrIS."*

8. *Lines 87-89: As I mentioned above, these references should be properly presented in the introduction.*

See response to comment above.

9. *Line 92: I might say "a main source" rather than "the main source". The branch of Akuliarusiarsuup Kuua that drains Russell Glacier and the lateral margin of Isunnguata Sermia is also pretty significant.*

Changed; however worth noting that the other branch (Russell and lateral margin of IS), whilst also a significant source, is probably at most ~ 4 times smaller than the contribution from LG, and likely much less at peak melt than LG, consistent with the overall drainage catchment at LG being about an order of magnitude larger than the Russell/Point 660 catchment (e.g. see discharge and catchment comparisons in Hawkings et al. (2021)).

10. *Line 150: Do you have any sense of the quantity of glacial surface runoff that makes it directly into the proglacial stream at Leverett Glacier? This could, in principle, be significant and seasonally variable.*

During most of the sampling period, we estimate this should be at least an order of magnitude smaller (but likely less), given that glacial-marginal runoff was relatively small based on visual observations (at least during the melt-seasons presented here). Direct surface runoff (i.e. not re-routed subglacially) might be slightly more important either very early or very late in the melt season; however very early time points here (in 2015) or late (2017) were sampled either via boreholes, upwellings, or immediately next to the glacier front (<20 m) and therefore should not have been significantly affected by "direct surface runoff" (sampling locations and distance from the ice margin are detailed in Table S2).

11. *Lines 237-238: This seems more a discussion type statement than a results heading.*

See response below

12. *Lines 237-296: I wonder if this should be results at all; it's mostly an elaborate description of the site context. Maybe move forward and give it its own heading. Or incorporate into an extended section 2.2 of the methods.*

This section had been moved to the results section due to a previous draft and confusion from reviewers that suggested moving into the results or discussion sections. We agree with the reviewer that this section better fits into the methods and will therefore move it back. We will rename its heading to *"2.8 Defining hydrological periods (water sourcing) based on the hydrological evolution of the LG proglacial river"* and refer to this new 2.8 section when first describing the hydrology in section 2.2. By re-structuring the methods section, we will now move Table 1 to the supplementary material.

13. *Line 263: I might consider a different term than "outburst period". To me, "outburst" implies something like a lake drainage event, whereas this is the normal spring escalation of the glaciohydrologic system.*

Most (if not all) of these pulses should indeed have been triggered by rapid lake drainage events on the glacier surfaces (as has been shown in previous melt season – e.g. Bartholomew et al. (2011). We will explicitly mention supraglacial lake drainage in the revised manuscript. Regarding the use of "outbusrt period", we choose to be consistent with previous reports on the same site and melt-season that used this terminology to explain this hydrological and hydrochemical perturbations and behaviours during that period (e.g. Hawkings et al., 2021; Kellerman et al., 2020; Lamarche-Gagnon et al., 2019; Hatton et al., 2019; Kohler et al., 2017)

*14. Line 319: A section header is very much missing here! This is no longer about "hydrological evolution".*

The section header was present on Ln 307 "*3.2 Microbial diversity and composition*". But we see how the TC preformatting of the preprint makes this a bit unclear. We apologise for the confusion (as indeed is definitely no longer about hydrological evolution). Given the restructuration of the results section due to previous comments, we will make sure section headers are updated accordingly.

*15. Figure 3: Is there a good reason for only plotting the top 5 here? I would be interested in tracing all of them, assuming the data is usable / significant. I also don't believe the data underlying this figure is in the supplemental tables (unless I some how missed it). Supplemental table 5 has a list of relative abundance of each of the OTUs, but I think in total for the whole suite, not by sample.*

See detailed response to Reviewer 1; also regarding corrections. In short this was due to avoid overplotting as representing all of the OTUs (1000s; or > dozens for the most abundant) would unfortunately be impossible on such a Figure. We can include all of the information (relative abundance of each OTU per sample) relating to Figure 3 as Supplementary information.

*16. Supplementary Tables: I found it difficult to understand the supplemental tables. The names of the sheets don't match the labels on the tabs. Some of the data is not legible without expanding the columns, and the meaning of some of the headings is not entirely clear.*

We are not entirely clear of which Tables the reviewer is referring to. Each supplementary table only has a single tab; perhaps the tab label on Table S2 which was referring to "2023 metadata"? We have removed the tab name. As for the column headers, most supplementary tables are meant as spreadsheets as they would be too wide and contain too much information to fit into a single page.

*17. Line 397-400: What about non-subglacially routed surface water? Did you do anything to look at the amount of water coming off the surface or off the ice-cored lateral moraines? This might be especially relevant before subglacially routed flow picks up.*

See above comment regarding non-subglacially routed surface waters, which we do not consider significant. We are not entirely certain as to which lateral moraine exactly the reviewer is referring to. However, we do not have discharge measurements from the small lateral streams, but, again, we consider those to only have a very minor contribution to the proglacial river during the years sampled.

*18. Line 409: Again, I don't particularly like the framing of these as "outburst events". You could just call them "spring events", like you do on the 2018 panel of figure 1.*

As per comment above, the use of "outburst events" is for consistency with previous studies describing the same events, especially in terms of hydrochemical context (see comment above). We feel that the more specific term "spring event" has stronger and more precise glaciological meanings, linking glacial velocity increases to meltwater deliveries in distributed drainage channels early in the melt-season. We therefore limit the use of "spring event" here to the very first hydrological pulse (outburst) of the melt-seasons; note our use of the phrase "putative spring event" (Ln 288) given our lack of data for that event in 2018. We also refer to the study from Bartholomew et al. (2011) which describes in detail these hydrological pulses (outbursts) and how the very first pulse (outburst) of the season at Leverett Glacier is likely "[…] *the result of initial access of meltwater to the subglacial drainage system through moulins and crevasses low down on the glacier following the onset of spring melting.*" and may differ from subsequent pulses/outbursts (see Bartholomew et al. 2011).

19. *Line 414: Is lysobacter really a "putative anaerobic" taxon?*

*Lysobacter* specifically is not necessarily an anaerobic taxa. Ln 414 referred to relative increase in relative abundances of putative anaerobic taxa compared to other hydrological periods, not necessarily that all taxa identified or highlighted during these periods to be putative anaerobes. We have re-phrased and re-formatted this paragraph to address comment 12 of reviewer 1 above and hope that this will make the statement less ambiguous. This particular sentence will be modified to:

*"These geochemical patterns are consistent with the increase in the relative abundance of taxa normally associated to sediment and/or anoxic environments exported to the ice margin during that period (Fig. 4c; Table S4-S5), including methanogens".*

20. *Lines 419-421: I'm not entirely convinced of this statement. What makes this period unique is the abundance of OTU 3, 4, and 6 (at least on Fig 3). To me these seem more like soil bacteria than a deep anaerobic source. Again, surface runoff could have been a more significant component in the early season. Whatever the source, it's a supply that is entirely or nearly entirely absent in different conditions.*

Here, we mostly based our interpretations from Fig. 4c and Table S4 and S5 (as referenced in the text), as they contain more information than Fig 3b (e.g. OTUs 11, 20, 23, etc.). We do not necessarily disagree with the reviewer that OTUs 3, 4, 6 are not necessarily indicative of strict anoxic environments, and indeed can be related to more cosmopolitan bacteria from soil or groundwater habitats. However these environments we feel would be more similar to the subglacial rather than to the surface environments (e.g. overridden paleosols). Ln 419-420 do not allude specifically to "deep anoxic sources" but instead highlights higher degree of heterogeneity in source environments: *"[…] the mobilisation of subglacial material from more heterogeneous sources earlier in the melt season and during outburst events".* We feel that these messages are more clear following our re-structuration of this paragraph (e.g. see above comment).

We also want to highlight a proposed revised comment in section 4.2 relating to pre-melt borehole samples and smaller lateral (marginal) outlets where we also highlight potential contribution of soil, groundwater, and potentially surface influence (will be added to Ln 499-500 of initial manuscript):

*"[...] Near identical methanotrophic (and methylotrophic) populations were also overrepresented in small methane-rich subglacial outflows of the same GrIS sector throughout the 2012, 2018 and 2019 melt seasons (Dieser et al., 2014; Znamínko et al., 2023). Unlike for the LG river here, however, these relatively smaller lateral meltwater outlets drain much smaller areas of the bed constrained near the ice margin (Hawkings et al., 2021) and might not be representative of ice sheet environments at large; e.g. bear a stronger imprint from more recently overlain material and/or from soil-percolated groundwaters (Graly et al., 2017), as well as influence from surface-derived organic matter (e.g. Andrews et al. (2018))."*

21. *Lines 436-440: This list doesn't match with the late melt highlighted taxa in fig 4. The OTUs should be 2,5,8,13, and 25 (i.e. not 1 and 4, which do not appear to increase here, but 8 and 13 instead). And again, I'm not entirely convinced of the diagnosis of this as a fingerprint for the ice sheet surface.*

The reviewer is correct that OTUs 2, 5, 8, 13 correspond to top OTUs highlighted by LEfSe for the 2017 late-melt periods. However, this paragraph is not solely discussing the late-melt 2017 samples, but also samples collected during the 2015 peak-flow period following the outburst period (see Ln 425-430 of the initial manuscript), where, for example, OTU4 is also highlighted on Figure 4c. As for OTU1, even though was present at a larger relative abundance in the 2018 samples, its relative abundance amongst the 2015 samples was highest during peak-flow and higher in the late-melt 2017 samples than in most of the 2015 samples (Fig. 3b, 4c).

As for these above OTUs fingerprinting surface environments, we also agree that they do not necessarily and only point towards this conclusion, nor that we make such a claim in the manuscript. The previous sentence (Ln 433-435 of the initial manuscript) states: *"we can expect a higher proportion of exported microbial assemblages to be derived from glacial surfaces, as well as to consist of more stable and widespread (sub)glacial populations (e.g. putative 435 generalists; Dubnick et al., 2023)".* That is these results point towards both a potential relative increase in a surface signal and/or a decrease in populations related to more heterogeneous subglacial sedimentary environments towards streamlining of microbial assemblages that are made up of relatively more generalist (sub)glacial populations (e.g. *Rhodoferax* OTU1).

As mentioned in the manuscript, we do not claim nor expect a clear-cut signal between different source environments, and we understand and agree that overall, proglacial runoff samples mostly contain OTUs related to the subglacial environment (e.g. explicitly stated on Ln 483-486)). This is also why all sampled periods still bear a strong "subglacial signal" - e.g. OTU 8, 13, 19 identified by LEfSe during peak flow and late melt (Fig. 4c, Table S4-5), but we still believe that relative changes in relative abundance of certain populations (those highlighted in the paper) alongside the overall glacio-hydrological context can inform on potential mixing and sourcing environments.

That being said, we do see how Ln436-440 can be improved to carry the above message, and we will better deconvolute and separate the references related to the different OTUs listed and their significance.

This section will now read:

*"Therefore, once the subglacial environment has been extensively flushed and drainage system established and stabilised, we can expect a higher proportion of exported microbial assemblages to be derived from glacial surfaces, as well as to consist of more stable and widespread cosmopolitan (sub)glacial populations (e.g. putative generalists; Dubnick et al., 2023). This is reflected here by a relative increase during these periods in OTUs related to bacterial populations commonly found on glacial surfaces (e.g. OTUs 4 (Microbacteriaceae), 5*

*(Luteolibacter), 25 (Pedobacter); Bradley et al. (2023) , Rathore et al. (2022) , Zhang et al. (2012)) and/or reported in (sub)glacial systems more broadly and consitently (e.g. OTUs 1 (Rhodoferax) and 2 (Polaromonas); e.g Kohler et al. (2020), Doyle and Christner (2022), Dubnick et al. (2023); Fig. 4, Table S5)."*

22. *Line 452: Typo: "appart"*

Changed

23. *Lines 478-483:  This might cast some shade on the notion that the "2017 late-melt" is a surface signal. Or maybe OTU 7 has some very particular circumstances under which it flourishes.*

    This section will be re-structed based on our answer to reviewer 1 (comment 14) above and should also address this point.

24. *Line 492: Is there a figure limit in this journal?  S3 seems quite important, so I would make it part of the manuscript proper if allowed.*

Thank you we will consider moving the figure to the main document.

References cited in our response to reviewer

Andrews, M. G., Jacobson, A. D., Osburn, M. R., and Flynn, T. M.: Dissolved Carbon Dynamics in Meltwaters From the Russell Glacier, Greenland Ice Sheet, Journal of Geophysical Research: Biogeosciences, 123, 2922-2940, 10.1029/2018JG004458, 2018.

Bartholomew, I., Nienow, P., Sole, A., Mair, D., Cowton, T., Palmer, S., and Wadham, J.: Supraglacial forcing of subglacial drainage in the ablation zone of the Greenland ice sheet, Geophysical Research Letters, 38, n/a-n/a, 10.1029/2011GL047063, 2011.

Bourquin, M., Busi, S. B., Fodelianakis, S., Peter, H., Washburne, A., Kohler, T. J., Ezzat, L., Michoud, G., Wilmes, P., and Battin, T. J.: The microbiome of cryospheric ecosystems, Nature Communications, 13, 3087, 10.1038/s41467-022-30816-4, 2022.

Boyd, E. S., Skidmore, M., Mitchell, A. C., Bakermans, C., and Peters, J. W.: Methanogenesis in subglacial sediments, Environ Microbiol Rep, 2, 685-692, 10.1111/j.1758-2229.2010.00162.x, 2010.

Bradley, J. A., Trivedi, C. B., Winkel, M., Mourot, R., Lutz, S., Larose, C., Keuschnig, C., Doting, E., Halbach, L., Zervas, A., Anesio, A. M., and Benning, L. G.: Active and dormant microorganisms on glacier surfaces, Geobiology, 21, 244-261, https://doi.org/10.1111/gbi.12535, 2023.

Cameron, K. A., Stibal, M., Hawkings, J. R., Mikkelsen, A. B., Telling, J., Kohler, T. J., Gözdereliler, E., Zarsky, J. D., Wadham, J. L., and Jacobsen, C. S.: Meltwater export of prokaryotic cells from the Greenland ice sheet, Environmental Microbiology, 19, 524-534, 10.1111/1462-2920.13483, 2017.

Christiansen, J. R. and Jørgensen, C. J.: First observation of direct methane emission to the atmosphere from the subglacial domain of the Greenland Ice Sheet, Scientific Reports, 8, 16623, 10.1038/s41598-018-35054-7, 2018.

Christner, B. C., Priscu, J. C., Achberger, A. M., Barbante, C., Carter, S. P., Christianson, K., Michaud, A. B., Mikucki, J. A., Mitchell, A. C., Skidmore, M. L., Vick-Majors, T. J., and the, W. S. T.: A microbial ecosystem beneath the West Antarctic ice sheet, Nature, 512, 310-313, 10.1038/nature13667, 2014.

Davis, C. L., Venturelli, R. A., Michaud, A. B., Hawkings, J. R., Achberger, A. M., Vick-Majors, T. J., Rosenheim, B. E., Dore, J. E., Steigmeyer, A., Skidmore, M. L., Barker, J. D., Benning, L. G., Siegfried, M. R., Priscu, J. C., Christner, B. C., Barbante, C., Bowling, M., Burnett, J., Campbell, T., Collins, B., Dean, C., Duling, D., Fricker, H. A., Gagnon, A., Gardner, C., Gibson, D., Gustafson, C., Harwood, D., Kalin, J., Kasic, K., Kim, O.-S., Krula, E., Leventer, A., Li, W., Lyons, W. B., McGill, P., McManis, J., McPike, D., Mironov, A., Patterson, M., Roberts, G., Rot, J., Trainor, C., Tranter, M., Winans, J., Zook, B., and the, S. S. T.: Biogeochemical and historical drivers of microbial community composition and structure in sediments from Mercer Subglacial Lake, West Antarctica, ISME Communications, 3, 8, 10.1038/s43705-023-00216-w, 2023.

Dieser, M., Broemsen, E. L., Cameron, K. A., King, G. M., Achberger, A., Choquette, K., Hagedorn, B., Sletten, R., Junge, K., and Christner, B. C.: Molecular and biogeochemical evidence for methane cycling beneath the western margin of the Greenland Ice Sheet, ISME J, 8, 2305-2316, 10.1038/ismej.2014.59, 2014.

Doyle, S. M. and Christner, B. C.: Variation in bacterial composition, diversity, and activity across different subglacial basal ice types, The Cryosphere, 16, 4033-4051, 10.5194/tc-16-4033-2022, 2022.

Dubnick, A., Kazemi, S., Sharp, M., Wadham, J., Hawkings, J., Beaton, A., and Lanoil, B.: Hydrological controls on glacially exported microbial assemblages, Journal of Geophysical Research: Biogeosciences, 122, 1049-1061, 10.1002/2016JG003685, 2017.

Dubnick, A. J., Spietz, R. L., Danielson, B. D., Skidmore, M. L., Boyd, E. S., Burgess, D., Dhoonmoon, C., and Sharp, M.: Biogeochemical evolution of ponded meltwater in a High Arctic subglacial tunnel, The Cryosphere, 17, 2993-3012, 10.5194/tc-17-2993-2023, 2023.

Foreman, C. M., Sattler, B., Mikucki, J. A., Porazinska, D. L., and Priscu, J. C.: Metabolic activity and diversity of cryoconites in the Taylor Valley, Antarctica, Journal of Geophysical Research: Biogeosciences, 112, https://doi.org/10.1029/2006JG000358, 2007.

Graly, J. A., Drever, J. I., and Humphrey, N. F.: Calculating the balance between atmospheric CO2 drawdown and organic carbon oxidation in subglacial hydrochemical systems, Global Biogeochemical Cycles, 31, 709-727, 10.1002/2016gb005425, 2017.

Hamilton, T. L., Peters, J. W., Skidmore, M. L., and Boyd, E. S.: Molecular evidence for an active endogenous microbiome beneath glacial ice, ISME J, 7, 1402-1412, 10.1038/ismej.2013.31, 2013.

Hatton, J. E., Hendry, K. R., Hawkings, J. R., Wadham, J. L., Kohler, T. J., Stibal, M., Beaton, A. D., Bagshaw, E. A., and Telling, J.: Investigation of subglacial weathering under the Greenland Ice Sheet using silicon isotopes, Geochimica et Cosmochimica Acta, https://doi.org/10.1016/j.gca.2018.12.033, 2019.

Hawkings, J. R., Linhoff, B. S., Wadham, J. L., Stibal, M., Lamborg, C. H., Carling, G. T., Lamarche-Gagnon, G., Kohler, T. J., Ward, R., Hendry, K. R., Falteisek, L., Kellerman, A. M., Cameron, K. A., Hatton, J. E., Tingey, S., Holt, A. D., Vinšová, P., Hofer, S., Bulínová, M., Větrovský, T., Meire, L., and Spencer, R. G. M.: Large subglacial source of mercury from the southwestern margin of the Greenland Ice Sheet, Nature Geoscience, 10.1038/s41561-021-00753-w, 2021.

Hodson, A., Anesio, A. M., Tranter, M., Fountain, A., Osborn, M., Priscu, J., Laybourn-Parry, J., and Sattler, B.: Glacial ecosystems, Ecological Monographs, 78, 41-67, 10.1890/07-0187.1, 2008.

Kellerman, A. M., Hawkings, J. R., Wadham, J. L., Kohler, T. J., Stibal, M., Grater, E., Marshall, M., Hatton, J. E., Beaton, A., and Spencer, R. G. M.: Glacier Outflow Dissolved

Organic Matter as a Window Into Seasonally Changing Carbon Sources: Leverett Glacier, Greenland, Journal of Geophysical Research: Biogeosciences, 125, e2019JG005161, 10.1029/2019jg005161, 2020.

Kohler, T. J., Žárský, J. D., Yde, J. C., Lamarche-Gagnon, G., Hawkings, J. R., Tedstone, A. J., Wadham, J. L., Box, J. E., Beaton, A. D., and Stibal, M.: Carbon dating reveals a seasonal progression in the source of particulate organic carbon exported from the Greenland Ice Sheet, Geophysical Research Letters, 6209-6217, 10.1002/2017GL073219, 2017.

Kohler, T. J., Vinšová, P., Falteisek, L., Žárský, J. D., Yde, J. C., Hatton, J. E., Hawkings, J. R., Lamarche-Gagnon, G., Hood, E., Cameron, K. A., and Stibal, M.: Patterns in Microbial Assemblages Exported From the Meltwater of Arctic and Sub-Arctic Glaciers, Frontiers in Microbiology, 11, 10.3389/fmicb.2020.00669, 2020.

Lamarche-Gagnon, G., Wadham, J. L., Sherwood Lollar, B., Arndt, S., Fietzek, P., Beaton, A. D., Tedstone, A. J., Telling, J., Bagshaw, E. A., Hawkings, J. R., Kohler, T. J., Zarsky, J. D., Mowlem, M. C., Anesio, A. M., and Stibal, M.: Greenland melt drives continuous export of methane from the ice-sheet bed, Nature, 565, 73-77, 10.1038/s41586-018-0800-0, 2019.

Linhoff, B. S., Charette, M. A., Nienow, P. W., Wadham, J. L., Tedstone, A. J., and Cowton, T.: Utility of 222Rn as a passive tracer of subglacial distributed system drainage, Earth and Planetary Science Letters, 462, 180-188, http://dx.doi.org/10.1016/j.epsl.2016.12.039, 2017.

Michaud, A. B., Dore, J. E., Achberger, A. M., Christner, B. C., Mitchell, A. C., Skidmore, M. L., Vick-Majors, T. J., and Priscu, J. C.: Microbial oxidation as a methane sink beneath the West Antarctic Ice Sheet, Nature Geosci, 10, 582-586, 10.1038/ngeo2992, 2017.

Pain, A. J., Martin, J. B., and Young, C. R.: Sources and sinks of CO2 and CH4 in siliciclastic subterranean estuaries, Limnology and Oceanography, 64, 1500-1514, 10.1002/lno.11131, 2019.

Poniecka, E. A., Bagshaw, E. A., Sass, H., Segar, A., Webster, G., Williamson, C., Anesio, A. M., and Tranter, M.: Physiological Capabilities of Cryoconite Hole Microorganisms, Frontiers in Microbiology, 11, 10.3389/fmicb.2020.01783, 2020.

Rathore, M., Sinha, R. K., Venkatachalam, S., and Krishnan, K. P.: Microbial diversity and associated metabolic potential in the supraglacial habitat of a fast-retreating glacier: a case study of Patsio glacier, North-western Himalaya, Environmental Microbiology Reports, 14, 443-452, https://doi.org/10.1111/1758-2229.13017, 2022.

Sáenz, J. S., Roldan, F., Junca, H., and Arbeli, Z.: Effect of the extraction and purification of soil DNA and pooling of PCR amplification products on the description of bacterial and archaeal communities, Journal of Applied Microbiology, 126, 1454-1467, 10.1111/jam.14231, 2019.

Sharp, M., Parkes, J., Cragg, B., Fairchild, I. J., Lamb, H., and Tranter, M.: Widespread bacterial populations at glacier beds and their relationship to rock weathering and carbon cycling, Geology, 27, 107-110, 10.1130/0091-7613(1999)027<0107:wbpagb>2.3.co;2, 1999.

Stevens, I. T., Irvine-Fynn, T. D. L., Edwards, A., Mitchell, A. C., Cook, J. M., Porter, P. R., Holt, T. O., Huss, M., Fettweis, X., Moorman, B. J., Sattler, B., and Hodson, A. J.: Spatially consistent microbial biomass and future cellular carbon release from melting Northern Hemisphere glacier surfaces, Communications Earth & Environment, 3, 275, 10.1038/s43247-022-00609-0, 2022.

Stibal, M., Wadham, J. L., Lis, G. P., Telling, J., Pancost, R. D., Dubnick, A., Sharp, M. J., Lawson, E. C., Butler, C. E. H., Hasan, F., Tranter, M., and Anesio, A. M.: Methanogenic potential of Arctic and Antarctic subglacial environments with contrasting organic carbon sources, Global Change Biology, 18, 3332-3345, 10.1111/j.1365-2486.2012.02763.x, 2012.

Wadham, J. L., Hawkings, J. R., Tarasov, L., Gregoire, L. J., Spencer, R. G. M., Gutjahr, M., Ridgwell, A., and Kohfeld, K. E.: Ice sheets matter for the global carbon cycle, Nature Communications, 10, 3567, 10.1038/s41467-019-11394-4, 2019.

Xiang, S.-R., Shang, T.-C., Chen, Y., Jing, Z.-F., and Yao, T.: Dominant Bacteria and Biomass in the Kuytun 51 Glacier, Applied and Environmental Microbiology, 75, 7287-7290, doi:10.1128/AEM.00915-09, 2009.

Zawierucha, K., Trzebny, A., Buda, J., Bagshaw, E., Franzetti, A., Dabert, M., and Ambrosini, R.: Trophic and symbiotic links between obligate-glacier water bears (Tardigrada) and cryoconite microorganisms, PLOS ONE, 17, e0262039, 10.1371/journal.pone.0262039, 2022.

Zhang, D.-C., Wang, H.-X., Liu, H.-C., Dong, X.-Z., and Zhou, P.-J.: Flavobacterium glaciei sp. nov., a novel psychrophilic bacterium isolated from the China No.1 glacier, International Journal of Systematic and Evolutionary Microbiology, 56, 2921-2925, https://doi.org/10.1099/ijs.0.64564-0, 2006.

Zhang, W., Zhang, G., Liu, G., Li, Z., Chen, T., and An, L.: Diversity of Bacterial Communities in the Snowcover at Tianshan Number 1 Glacier and its Relation to Climate and Environment, Geomicrobiology Journal, 29, 459-469, 10.1080/01490451.2011.581329, 2012.

Zhu, F., Wang, S., and Zhou, P.: Flavobacterium xinjiangense sp. nov. and Flavobacterium omnivorum sp. nov., novel psychrophiles from the China No. 1 glacier, International Journal of Systematic and Evolutionary Microbiology, 53, 853-857, https://doi.org/10.1099/ijs.0.02310-0, 2003.

Znamínko, M., Falteisek, L., Vrbická, K., Klímová, P., Christiansen, J. R., Jørgensen, C. J., and Stibal, M.: Methylotrophic Communities Associated with a Greenland Ice Sheet Methane Release Hotspot, Microbial Ecology, 10.1007/s00248-023-02302-x, 2023.